# Deep-branching Chloroflexota lineages illuminate the eco-evolutionary foundation of cross-ecosystem colonization

Lucas Serra Moncadas ◍ , Alisa Shakurova, Cyrill Hofer ◍ & Adrian-Stefan Andrei ◍ ✉

Bacteria span Earth's ecosystems, coupling ecological versatility with genome-architectural reconfiguration across shifting physicochemical conditions. Yet the genomic routes by which free-living lineages cross ecosystem boundaries, and the consequences for genome architecture, remain poorly understood. Here, we use comparative and evolutionary genomics to investigate a soil-to-sediment-to-freshwater transition in Limnocylindria, an abundant clade within the Chloroflexota phylum. Two sister families show contrasting strategies. CSP1-4 expands genomes through niche-specific gene acquisition, whereas Limnocylindraceae undergoes genome reduction and metabolic simplification—revealing alternative evolutionary routes to similar ecological outcomes. In Limnocylindraceae, the loss of key DNA glycosylases coincides with degradation of base excision repair and is consistent with a hypermutator state that may have accelerated genomic erosion during freshwater specialization, potentially facilitating ecological expansion. This reductive genome trajectory is associated with a freshwater-adapted lineage with unexpectedly high GC content, challenging canonical links between base composition and genome size. While mutational processes appear to dominate genome restructuring, proteome-level patterns suggest selection favoring carbon- and nitrogen-efficient amino acid usage, implying that adaptive refinement can emerge alongside primarily non-adaptive dynamics. Overall, our findings are consistent with mutation-driven genome reduction and proteome optimization acting in concert to support cross-ecosystem boundary crossing and freshwater specialization in a free-living Chloroflexota lineage.

Life on Earth likely originated during the early Archean Eon[1,2]—approximately 4.0 to 3.6 billion years ago—a period marked by the formation of Earth's first continental crust and its primordial oceans[3]. Although the exact nature and habitat of life's origin[2] remain subjects of debate, early life forms swiftly diversified, adapting to and colonizing the newly sprung environments[4]. Today, Bacteria and Archaea (the closest living descendants of these primordial organisms) numerically dominate Earth's ecosystems, with an estimated $1.2 \times 10^{30}$ cells[5]. They are the most abundant and pervasive life forms, thriving in virtually every environment—from frozen polar caps and blazing hydrothermal vents to the abyssal depths of the ocean and the lofty heights of the atmosphere[6–9].

Bacteria are notoriously adaptable, encompassing vast metabolic versatility[10], high phylogenetic diversity[11], and the ability to inhabit a wide array of ecosystems[4]. This adaptability stems from their capacity to reshape their genomic architecture through mechanisms such as

Limnological Station, Department of Plant and Microbial Biology, University of Zurich, Kilchberg, Switzerland. ✉e-mail: stefan.andrei@limnol.uzh.ch

horizontal gene transfer[12], mutations[13], and genetic rearrangements[14]. These processes enable bacteria to swiftly acquire new traits and adapt to changing environmental conditions, facilitating the exploration of novel ecological niches and fostering diversification[15]. Although bacteria can expand into related niches by exploiting new resources or undergoing metabolic rewiring, the ways and mechanisms through which they perform large-scale ecosystem transitions—that is, shifts into fundamentally different environments—remain poorly charted[16].

Our primary understanding of how bacteria adapt to and colonize new environments stems from studies on pathogenic and parasitic lineages, which often originate from free-living environmental ancestors[17–19]. Such investigations have been instrumental in revealing how mechanisms like horizontal gene transfer, gene loss, duplication, and functional divergence enable bacteria to adapt to host, enhancing their virulence and survival[20]. However, transitions to host-associated habitats frequently involve mobile genetic element invasions[21,22] or degenerative processes associated with small population sizes, including genome shrinkage, pseudogenization, and metabolic impairment[20]. In contrast, the evolutionary dynamics of free-living species—characterized by larger population sizes and likely different selective pressures—remain insufficiently explored[23].

Ecosystem transitions in free-living bacteria represent rare yet pivotal evolutionary events that require intricate genomic innovation and ecological rewiring[16]. While relatively few lineages are known to breach ecosystem boundaries, several notable examples exist, including SAR11[24] (marine to freshwater), Actinomarinales[25] (freshwater to marine?), Salinispora[26] (terrestrial to marine), and Methylophilaceae[27] and Phycisphaerae[28] (sediment to freshwater). These transitions are typically characterized by genomic rearrangements, reductions in metabolic capacity[29], and the acquisition of habitat-specific gene sets[16]. Nevertheless, many aspects of the mechanisms driving these transformations and the eco-evolutionary processes influencing them remain unclear[29]. This information deficit is due not only to the limited number of documented cases but also to lineage extinctions over evolutionary timescales, which may have erased whole transitional clades[30]. Consequently, many crucial evolutionary events remain obscured, hindering our ability to fully comprehend the complexity of ecosystem-level transitions and their broader impact on bacterial ecology and evolution.

To address how microbial genomes evolve during ecosystem transitions, we analyse 236 metagenome-assembled genomes from Limnocylindria, an uncultivated, class-level lineage within the phylum Chloroflexota. This group spans soils, sediments, and freshwater environments. Within Limnocylindria, two sister families, CSP1-4 and Limnocylindraceae, exhibit contrasting adaptive strategies. CSP1-4 shows expansion into freshwater sediments and the water column, coupled to habitat-specialized gene acquisition, larger genomes, and signatures consistent with relaxed selection pressure. In contrast, Limnocylindraceae shows extensive genome reduction and metabolic simplification and is largely restricted to freshwater environments. This divergence provides a unique window into evolutionary alternatives for bacterial lineages crossing ecological boundaries, with one trajectory shaped by functional acquisition and the other by reductive specialization.

To dissect the reductive route in detail, we focus on Limnocylindraceae, a freshwater lineage whose unusually small yet GC-rich genome (median 1.307 Mbp; 63.15% GC) defies long-held assumptions about genome reduction and base composition in free-living bacteria[31,32]. By integrating phylogenomic reconstructions, gene-level selection analyses, and genome-informed metabolic reconstructions, we show that reductions in genome size and shifts in GC content are primarily driven by non-adaptive processes, likely exacerbated by the loss of DNA repair mechanisms. In parallel, adaptive signals emerge at the proteome level, where carbon and nitrogen economization suggests selective pressure to reduce biosynthetic costs. Together, these findings establish Limnocylindraceae as a model for how bacterial life can adapt and persist at the edge of genomic and metabolic minimalism while crossing ecosystem boundaries[31,32].

## Results and discussion

### Eco-evolutionary history

In this study, we expand upon our previously published pdCEL database[15], designed to explore the eco-evolutionary forces driving adaptation and speciation in freshwater bacteria. pdCEL encompasses 5500 metagenome-assembled genomes (MAGs) derived from time-series samples collected from lakes spanning a wide spectrum of trophic states, from oligotrophic to dystrophic.

To investigate the evolutionary history of the Limnocylindria class (Chloroflexota), we conduct a taxonomy-based retrieval, recovering 72 relevant MAGs from the pdCEL database (Supplementary Fig. S2). We further augment this dataset with 164 publicly available MAGs from the GTDB R220 release (as of June 2024). This expanded dataset increases taxonomic breadth and resolution, enabling more accurate phylogenetic inferences and a clearer reconstruction of the clade's evolutionary history (Fig. 1a). We built a rooted phylogenomic tree using maximum-likelihood methods based on 118 single-copy proteins, employing profile mixture models with the LG substitution matrix and 30 composition profiles to account for amino acid frequency variability and evolutionary rate heterogeneity across sites. This approach reveals four major phylogenetic clades, each with high support values (SH-aLRT = 100; UFBoot = 100), with the two evolutionary youngest forming a sister-group relationship (Fig. 1a).

We analyse the environmental distribution of the four major taxonomic clusters using the Sandpiper database, which employs SingleM to profile taxa from over 248,000 metagenomes, offering a comprehensive, high-resolution view of microbial diversity across diverse ecosystems[33] (see "Methods"). By integrating distribution and abundance patterns within a phylogenetic framework, we reveal a clear evolutionary trajectory marked by ecosystem-level habitat transitions.

The ancestral P2-11E lineage (1,352 samples) is predominantly soil-associated (94.5%), with a peak environmental abundance of 9%. Its descendant, QHBO01 (2441 samples), retains a strong preference for soil habitats (92.8%), where it reaches abundances of up to 30% (Fig. 1a). In contrast, the sister lineages CSP1-4 and Limnocylindraceae exhibit pronounced ecological shifts. CSP1-4 (9454 samples), while primarily found in soil (71.4%), has significantly expanded into sediments (25%), achieving comparable peak abundances of 18% in both environments (Fig. 1a). Limnocylindraceae (1352 samples), however, is almost exclusively freshwater-associated (98.7%)—presence reconfirmed through CARD-FISH (Supplementary Fig. S1)—with peak abundances of 11%, particularly in shallow, runoff-affected lakes (high-abundance habitats shown in green; Fig. 1 upper left inset). These patterns highlight a cross-ecosystem transition by Limnocylindraceae, which maintains a tenuous link to soil habitats by thriving in freshwater systems heavily influenced by terrestrial inputs (Fig. 1a).

### Evolution of genomic and metabolic architectures

We compiled a comprehensive genomic dataset for the Limnocylindria class, comprising genomes with a median completeness of 80% and contamination of 1.6% (n = 236). The pdCEL database significantly enhanced genomic representation, particularly for CSP1-4 (n = 5) and Limnocylindraceae (n = 67 genomes, representing five novel species), sincreasing the latter's genomic coverage by 68% and expanding its phylogenetic diversity by 48.14%. A detailed examination of genome architectures across the four major phylogenetic lineages reveals significant evolutionary divergence. The families P2-11E, QHBO01, and CSP1-4 exhibit comparable genomic features, including median genome sizes ranging from 2.4 to 3.2 Mbp and consistently high GC content between 68% and 68.6%. In stark contrast, Limnocylindraceae exhibits a dramatic genome reduction, with median genome sizes

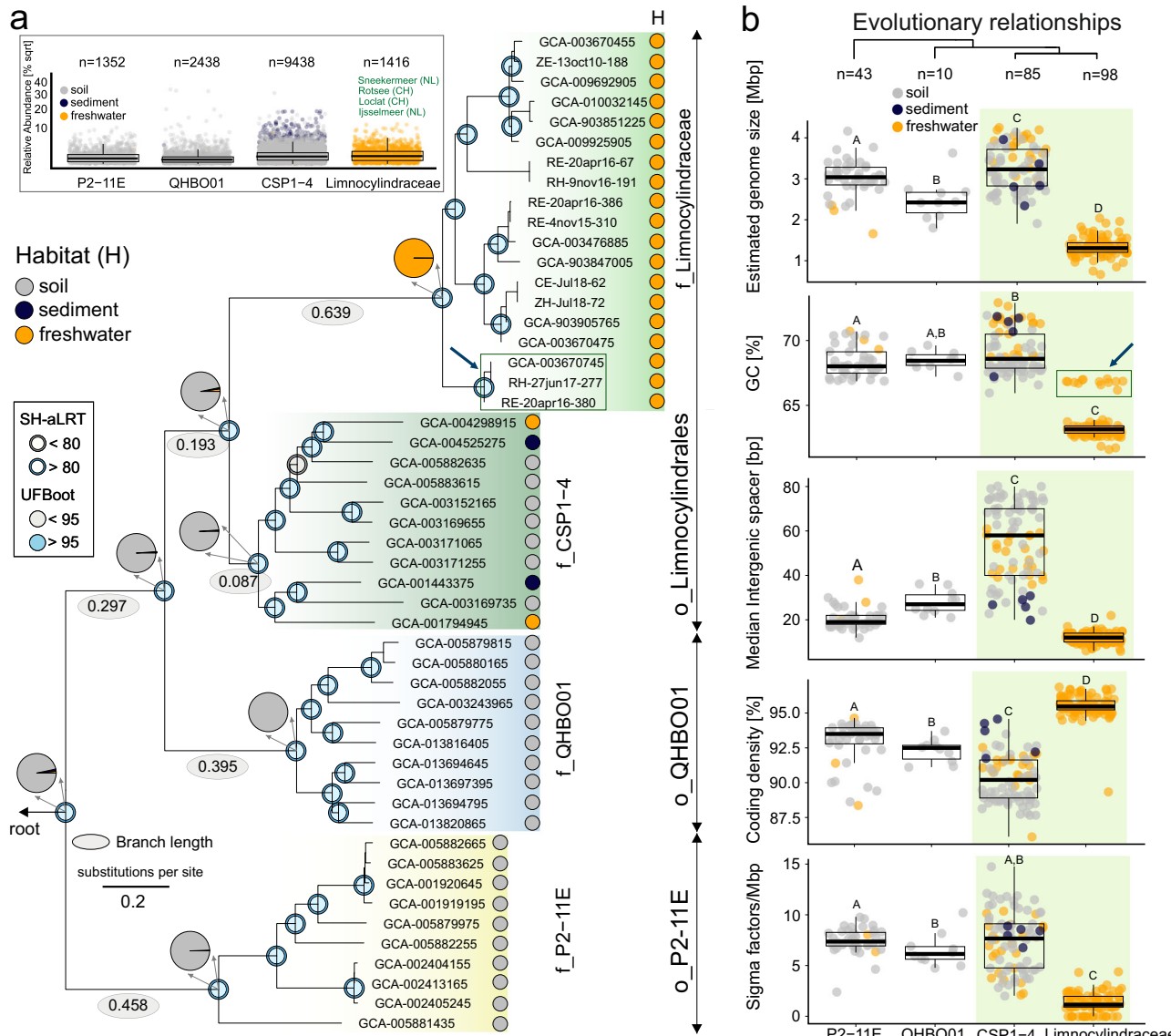

**Fig. 1 | Limnocylindria phylogenomics. a** Rooted maximum-likelihood phylogeny of the Limnocylindria class (Chloroflexota phylum) was constructed using the LG + C30 model, which combines the LG amino acid substitution model with a 30-profile mixture model to account for compositional heterogeneity. The inset (upper left panel) displays the distribution of the four taxonomic clades across worldwide microbiomes. Blue rings and circles denote approximate likelihood ratio test and Shimodaira-Hasegawa (SH-aLRT) and Ultrafast bootstrap (UFBoot) support values exceeding 80 and 95, respectively. Node insets represent pie charts. At each internal node, the inset pie chart shows the marginal posterior probability of each ancestral habitat occupancy (inferred under the Equal-Rates model with 1000 stochastic mappings); slice areas are proportional to these probabilities, and colours correspond to habitat categories. Tip labels are colored according to the habitat from which each genome was recovered. Grey ovals indicate branch length values. Scale bars represent the number of substitutions per site. The blue arrow indicates the ancestral Limnocylindraceae lineage. Outgroup rooting was

performed by using members of the Eremiobacterota phylum. **b** Breakdown of family-level genomic properties and features ordered in a phylogenetic fashion. The CSP1-4 and Limnocylindraceae sister lineages are highlighted by a green background. The blue arrows (panels **a** and **b**) highlight the phylogenetic position and GC content of a selected Limnocylindraceae lineage. Pairwise statistical differences are highlighted via capital letters. Boxplots sharing the same capital letter indicate no significant difference ($p > 0.05$), while boxplots with different capital letters are significantly different ($p \leq 0.05$). The central line across the boxplots identifies the median, marking the dataset's midpoint. The box itself demarcates the interquartile range, extending from the first quartile to the third quartile, encapsulating the central 50% of the data. The whiskers project from the box to the furthest data points not categorized as outliers and show the spread of the main body of the dataset. All statistics are reported in Supplementary Data S14. Source data are provided as a Source Data file.

halving to 1.3 Mbp, accompanied by a slight decrease in GC content (median 63.15%) (Fig. 1b).

The CSP1-4 family comprises 16 genera and 51 strictly habitat-specific species—35 from soil, 4 from sediment, and 11 from freshwater. Genome-scale metabolic reconstructions, evaluated via a pairwise similarity matrix, reveal the greatest metabolic overlap between soil and sediment lineages (Fig. 2a, b). Soil genomes uniquely encoded enzymes for depolymerizing plant polysaccharides and polyols (e.g.,

sucrose phosphorylase, 6-phospho-β-glucosidase, inosose dehydratase), along with nutrient-stress regulators (cstA, phoH/phoL, and phoB), reflecting possible adaptation to complex carbon inputs and episodic carbon–phosphate scarcity (Supplementary Data S11). In contrast, sediment lineages harbor the catalytic α- and β-subunits of the nitrate reductase/nitrite oxidoreductase complex, a complete nitrate/nitrite ABC transporter, and the high-affinity cytochrome d terminal oxidase, consistent with respiration in oxygen-poor,

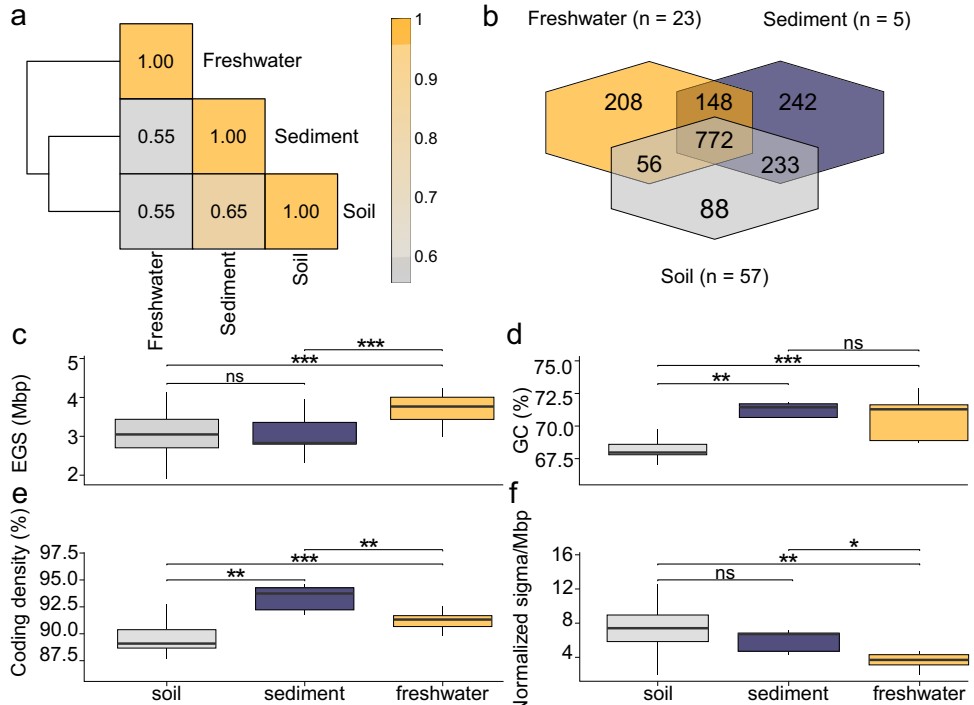

**Fig. 2 | Metabolic similarity and genomic features of CSP1-4 across ecosystems.** **a** Heatmap of Jaccard similarity values based on presence/absence of KEGG Orthology identifiers pooled by habitat (soil, sediment, freshwater) in CSP1-4 genomes. Colors range from grey (low similarity) to orange (high similarity); cell values indicate exact similarity scores. **b** Distribution of genome-encoded metabolic potential across habitats. Data are grouped by origin: soil (grey), sediment (purple), and freshwater (orange). **c**–**f** Boxplots showing variation in genome size, GC content, coding density, and normalized sigma factor counts across the three habitats. Pairwise significance is indicated by asterisks (*$p < 0.05$; **$p < 0.01$; ***$p < 0.001$, ns > 0.05); absence of stars denotes non-significant differences. The central line across the boxplots identifies the median, marking the dataset's midpoint. The box itself demarcates the interquartile range, extending from the first quartile to the third quartile, encapsulating the central 50% of the data. The whiskers project from the box to the furthest data points not categorized as outliers and show the spread of the main body of the dataset. All statistics are reported in Supplementary Data S14. Source data are provided as a Source Data file.

nitrate-rich sediments. Freshwater representatives are distinguished by expanded flagellar and chemotaxis loci—traits rare in Chloroflexota[34]—suggesting capacity for motility-driven navigation of heterogeneous aquatic niches (Supplementary Data S11). Genomic architecture differs consistently among habitats, reflecting distinct ecological constraints (Fig. 2c–f). Freshwater genomes have the largest estimated sizes and encode markedly fewer sigma factors (proxies for transcriptional regulatory complexity and ecological versatility) than those from soil or sediment, indicative of regulatory streamlining in planktonic environments (Fig. 2c). GC content and coding density were elevated in both sediment and freshwater lineages relative to soil representatives, suggesting habitat-specific selection for more compact and information-dense genomes outside terrestrial ecosystems. Together, these habitat-specific patterns in CSP1-4 gene content and genome structure support distinct ecological strategies shaped by carbon availability, nutrient dynamics, and redox conditions across terrestrial and aquatic environments (further details on Limnocylindria metabolism are provided in the Supplementary Note 1).

Striking genomic contrasts distinguish between the evolutionary younger sister families CSP1-4 and Limnocylindraceae (Fig. 1b). Limnocylindraceae exhibits significant genome size reduction, fewer sigma factors (key proteins essential for transcription initiation), and increased coding density resulting from reduced intergenic spacers (Fig. 1b, and Supplementary Fig. S3). While this genomic profile aligns with 'classical' models of genome reduction in free-living bacteria[23,35], it also notably deviates by its high GC content (median = 63.15%). To elucidate the evolutionary processes that may have given rise to this particular genomic architecture, we first examine the evolutionary context of its emergence.

Phylogenetic analyses reveal that Limnocylindraceae shares a common ancestry with the CSP1-4 clade (Fig. 1a). The latter occupies an exceptionally short branch stemming from its most recent common ancestor—the shortest in our phylogeny—indicating reduced evolutionary change from the ancestral state (0.087 substitutions per site since divergence). To further understand the evolutionary landscape that may have favored the emergence of Limnocylindraceae, we conduct a clade-specific selection force analysis on a set of approximately 800 genes conserved across the four clades. We observed that CSP1-4 exhibits a 60% increase in the number of genes under positive (diversifying) selection compared to its ancestral lineage (i.e., QHBO01), along with a 25% increase in the number of sites under positive selection (Supplementary Table S1). The elevated levels of positive selection observed in CSP1-4, the highest among the four clades, suggest that this lineage is undergoing adaptive evolution, characterized by an accumulation of non-synonymous substitutions. Additionally, the relaxation of selective pressure in CSP1-4 is complemented by an increase in transposase density, a higher number of redundant proteins, and larger intergenic spacer regions—the longest among the four clades (Fig. 1b, and Supplementary Fig. S4). Together, these factors indicate decreased negative (purifying) selection and an enhanced potential for adaptive evolution[15,36,37]. These observations correlate with the habitat expansion of CSP1-4 in sediment and aquatic ecosystems.

## GC dynamics
The extant genomic diversity of freshwater-adapted Limnocylindraceae is encompassed within a single genus-level clade, Limnocylindrus. The basal branch of this genus (blue arrow, Fig. 1) exhibits the

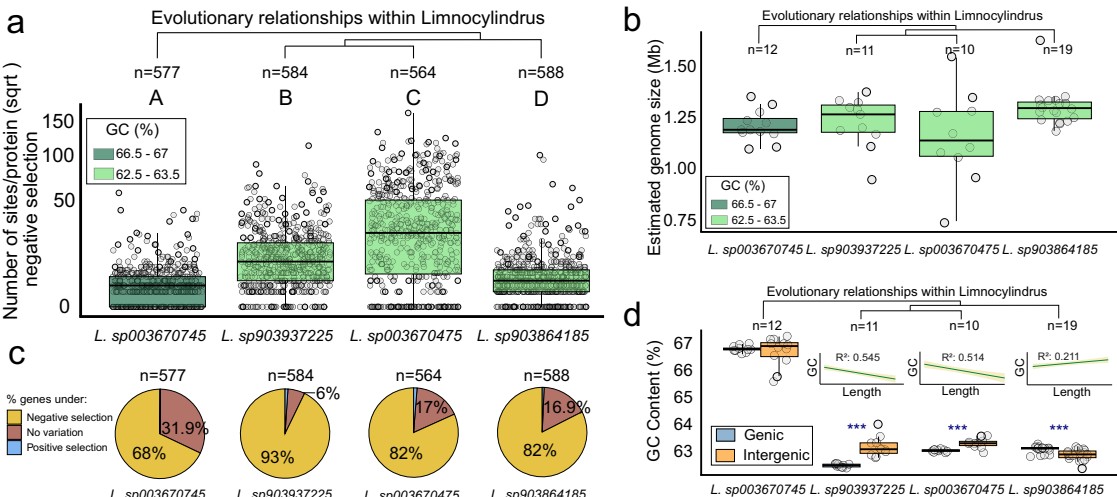

**Fig. 3 | Limnocylindrus selection force analyses. a** Genome-wide quantification of negatively selected sites per gene across four Limnocylindrus species, arranged in evolutionary order. **b** Genome size estimates for each species, with boxplot colors representing GC content. **c** Percentage distribution of genes under negative selection, positive selection, and classified as invariable within each species. **d** GC content distribution between genic (coding) and intergenic (non-coding) regions across the four species, with insets highlighting the relationships between GC content and the total length of genic and intergenic regions. Pairwise statistical differences are highlighted via the means of capital letters and stars (*$p < 0.05$; **$p < 0.01$; ***$p < 0.001$, ns ≤ 0.05). Boxplots sharing the same capital letter indicate no significant difference ($p > 0.05$), while boxplots with different capital letters are significantly different ($p ≤ 0.05$). The central line across the boxplots identifies the median, marking the dataset's midpoint. The box itself demarcates the interquartile range, extending from the first quartile to the third quartile, encapsulating the central 50% of the data. The whiskers project from the box to the furthest data points not categorized as outliers and show the spread of the main body of the dataset. The grey bands represent 95% confidence interval for the mean predicted by the regression. All statistics are reported in Supplementary Data S14. Source data are provided as a Source Data file.

defining features of the lineage: reduced genome size, high coding density, and low counts of sigma factors and signal transduction domains, consistent with other representatives (Fig. 3b, Supp. Fig. S5, and Supp. Data S1). However, it retains a significantly higher GC content of 66.8% ($n = 14$; Fig. 1b) than the younger species of the genus (median = 63.1%; $n = 85$). This suggests that genome size reduction in Limnocylindraceae likely preceded the decrease in GC content. Therefore, the existence of such a genus-level clade with uniform genomic features—except for variations in GC content—offers a unique opportunity to study the evolution of GC content independently of genome size.

Selection-pressure analyses of 52 MAGs representing four freshwater Limnocylindrus species—defined by >95% average nucleotide identity and >70% conserved DNA—indicate that the basal lineage contains both fewer genes and sites per gene under negative selection than its evolutionary younger, lower-GC relatives (Fig. 3a,c). Thus, the approximately 3% decrease in GC content within Limnocylindrus is associated with an increase in both the number of genes and the number of sites under negative selection, along with a reduction in the number of fully conserved (invariable) genes—defined here as genes with no detectable nucleotide substitutions across the alignment, consistent with strong purifying selection. No substantial differences in GC content are observed between species-specific genes and the shared core genome (Supplementary Fig. S13). This indicates that negative selection affects a larger portion of the genome, while a higher mutation rate introduces more sequence variation, as evidenced by the predominance of synonymous substitutions—nucleotide changes that do not alter the amino acid sequence and are typically retained under negative selection— in most genes (Fig. 3c). Additionally, the decrease in invariable genes suggests that mutations occur more frequently in lower-GC genomes, resulting in fewer unchanged genes. Consequently, the combination of an increased mutation frequency and strong selection pressure leads to more genes under negative selection and fewer entirely invariable ones.

A further comparison of GC content of genes and intergenic spacers revealed that the P2-11E, QHB001, and CSP1-4 main lineages exhibit higher GC content in their coding regions (i.e., genes) compared to their non-coding counterparts (Supplementary Fig. S6a). These findings are consistent with previous research indicating that non-coding regions tend to accumulate more mutations and experience weaker selection pressure than protein-coding regions, often leading to a lower GC content[38–40]. Notably, the family Limnocylindraceae showed higher GC content in the intergenic regions (Supplementary Fig. S6a). By examining four representative species, we found that in freshwater-adapted Limnocylindraceae, intergenic regions—despite their short median length (12 bp)—consistently exhibit higher GC content than coding regions (Fig. 3d). This pattern may reflect a reduced likelihood of mutation accumulation in these shorter sequences, as their limited length presents a smaller mutational target compared to longer coding regions. Overall, these findings suggest that the observed decrease in GC content within Limnocylindraceae is likely driven by the random accumulation of mutations. The selection pressure on the GC content of genes does not appear to be higher than that acting on intergenic spacers. Therefore, the differences in GC composition between the two are likely attributable to their overall size. Although GC content was not directly analysed in Bourguignon et al.[41], their finding that genome reduction in prokaryotes is frequently associated with elevated mutation rates supports a broader mutational framework. Our results align with this model and suggest that mutation-driven genome erosion may also influence base composition—manifesting as GC content reduction—in genome-reduced, free-living lineages such as Limnocylindraceae.

The codon frequency analyses of 20 proteogenic amino acids reveal that synonymous substitutions are replacing GC-rich codons with AT-rich ones, mirroring the overall decrease in genomic GC content (Fig. 4). In the freshwater Limnocylindrus, this substitution pattern is particularly pronounced, with 75% of cases exhibiting a distinctive "funnel-shaped" dynamic where ancestral high-GC codons are progressively replaced by codons with higher AT content (Fig. 4a,

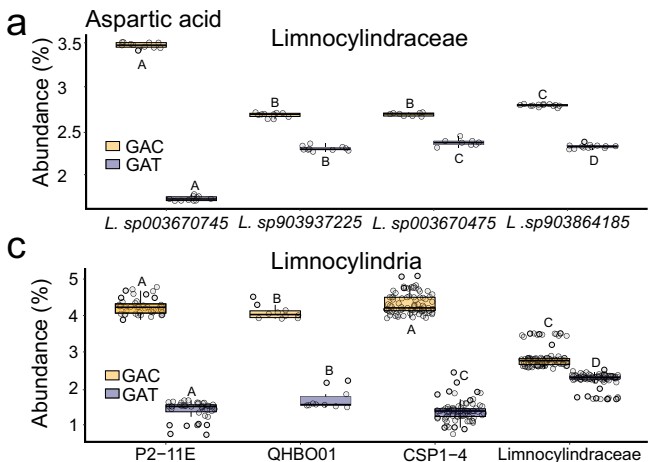

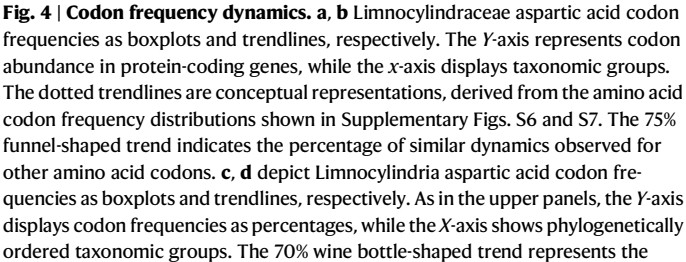

**Fig. 4 | Codon frequency dynamics. a, b** Limnocylindraceae aspartic acid codon frequencies as boxplots and trendlines, respectively. The *Y*-axis represents codon abundance in protein-coding genes, while the *x*-axis displays taxonomic groups. The dotted trendlines are conceptual representations, derived from the amino acid codon frequency distributions shown in Supplementary Figs. S6 and S7. The 75% funnel-shaped trend indicates the percentage of similar dynamics observed for other amino acid codons. **c, d** depict Limnocylindria aspartic acid codon frequencies as boxplots and trendlines, respectively. As in the upper panels, the *Y*-axis displays codon frequencies as percentages, while the *X*-axis shows phylogenetically ordered taxonomic groups. The 70% wine bottle-shaped trend represents the percentage of similar dynamics observed for other amino acid codons. Pairwise statistical differences are highlighted via the means of capital letters. Boxplots sharing the same capital letter indicate no significant difference ($p > 0.05$), while boxplots with different capital letters are significantly different ($p \leq 0.05$). The central line across the boxplots identifies the median, marking the dataset's midpoint. The box itself demarcates the interquartile range, extending from the first quartile to the third quartile, encapsulating the central 50% of the data. The whiskers project from the box to the furthest data points not categorized as outliers and show the spread of the main body of the dataset. All statistics are reported in Supplementary Data S14. Source data are provided as a Source Data file.

b, and Supplementary Fig. S8). To determine whether this trend is consistent across the entire Limnocylindria class, we performed codon frequency analyses on all major families and observed a similar pattern (Fig. 3b, d; and Supplementary Fig. S8). The evolutionary decrease in GC content is accompanied by a reduction in GC-rich codons and an increase in AT-rich ones, resulting in a "wine bottle"-shaped dynamic in 70% of the cases. This pattern showcases how the initial spread in the distribution of codon GC content (low frequency for AT-rich and high frequency for GC-rich) is followed by a narrowing in Limnocylindraceae resembling the shape of a wine bottle. Overall, these findings indicate that the decrease in GC content in Limnocylindraceae is likely driven by a shift in synonymous codon usage from GC-rich to AT-rich codons. This suggests a mutation-driven mechanism rather than ecological adaptation, as the number of sites under positive selection is too limited to explain the observed GC content changes.

## Emergence of reduced genomic architectures

Comparative genome-informed metabolic reconstructions reveal that genome shrinkage in Limnocylindraceae coincides with reductions in metabolic and biosynthetic capacities (Fig. 5, and Supplementary Note 2). Remarkably, this lineage lacks most DNA glycosylases involved in the base excision repair (BER) pathway (Fig. 5a). BER is an essential DNA repair mechanism that corrects small, non-helix-distorting base lesions arising from alkylation, deamination, oxidative damage, or spontaneous base loss[42]. We propose that the absence of these DNA glycosylases may underlie the distinctive genomic architecture observed in freshwater-adapted Limnocylindraceae and explore the consequences of their loss on GC content evolution.

The glycosylases Tag (K01246), AlkA (K01247), and MPG (K03652) play pivotal roles in excising alkylated bases such as 3-methyladenine (3-MeA) and 3-methylguanine (3-MeG), initiating BER. In bacteria lacking these enzymes, repair of alkylated bases is largely compromised. While 3-MeA lesions are highly cytotoxic due to their replication-blocking properties, they contribute minimally to mutagenesis as DNA polymerases rarely bypass them[43]. In contrast, 3-MeG lesions are less obstructive and can be misread during replication,

leading to mispairing with thymine instead of cytosine. This mispairing will eventually lead to G → A transition mutations[44]. The pronounced impact of 3-MeG-induced mutations on GC content stems from their higher DNA polymerase bypass frequency and consistent mispairing behavior[44,45].

Uracil DNA glycosylases Family 4 (TIGR03914), Ung (K03648) and Mug (K03649) excise uracil from DNA, preventing mutations from cytosine deamination or uracil misincorporation during replication. In the absence of these enzymes, uracil-induced lesions will most probably persist. Uracil arises via spontaneous deamination of cytosine, forming U·G mismatches, or through misincorporation of dUMP opposite adenine, forming U·A pairs. Unrepaired uracil in U·G mismatches is read as thymine during replication, leading to the insertion of adenine opposite uracil and resulting in G → A transition mutations[44,45]. Accumulation of these mutations further decreases genomic GC content as G·C pairs are replaced by A·T ones.

The enzymes Nei (K05522) and MutT (K03574) are crucial for repairing oxidative DNA damage. Nei excises oxidatively damaged pyrimidines like thymine glycol, preventing mispairing during replication[46]. MutT sanitizes the nucleotide pool by hydrolyzing 8-oxo-dGTP, an oxidized and mutagenic form of dGTP. Without MutT, 8-oxo-dGTP accumulates and is incorporated opposite adenine, forming A·8-oxo-G mispairs. This mispairing leads to A·T → C·G transversions[47]. Conversely, the absence of Nei allows the accumulation of oxidized pyrimidines, causing mispairing during replication and resulting in G·C → T·A transversions, which will decrease GC content[44].

Overall, we propose that the absence of key DNA glycosylases in Limnocylindraceae could lead to increased mutagenesis due to the accumulation of unrepaired DNA lesions, such as alkylated bases and deaminated cytosine residues. In the absence of repair, these lesions are likely to mispair during DNA replication, resulting in mutations. The elevated mutation frequency may promote the inactivation and subsequent deletion of non-essential or low-benefit gene sets, ultimately driving genome reduction through cumulative deletion events[48]. Moreover, the overall pattern of missing key enzymes in the BER pathway favors a decrease in GC content through G → A

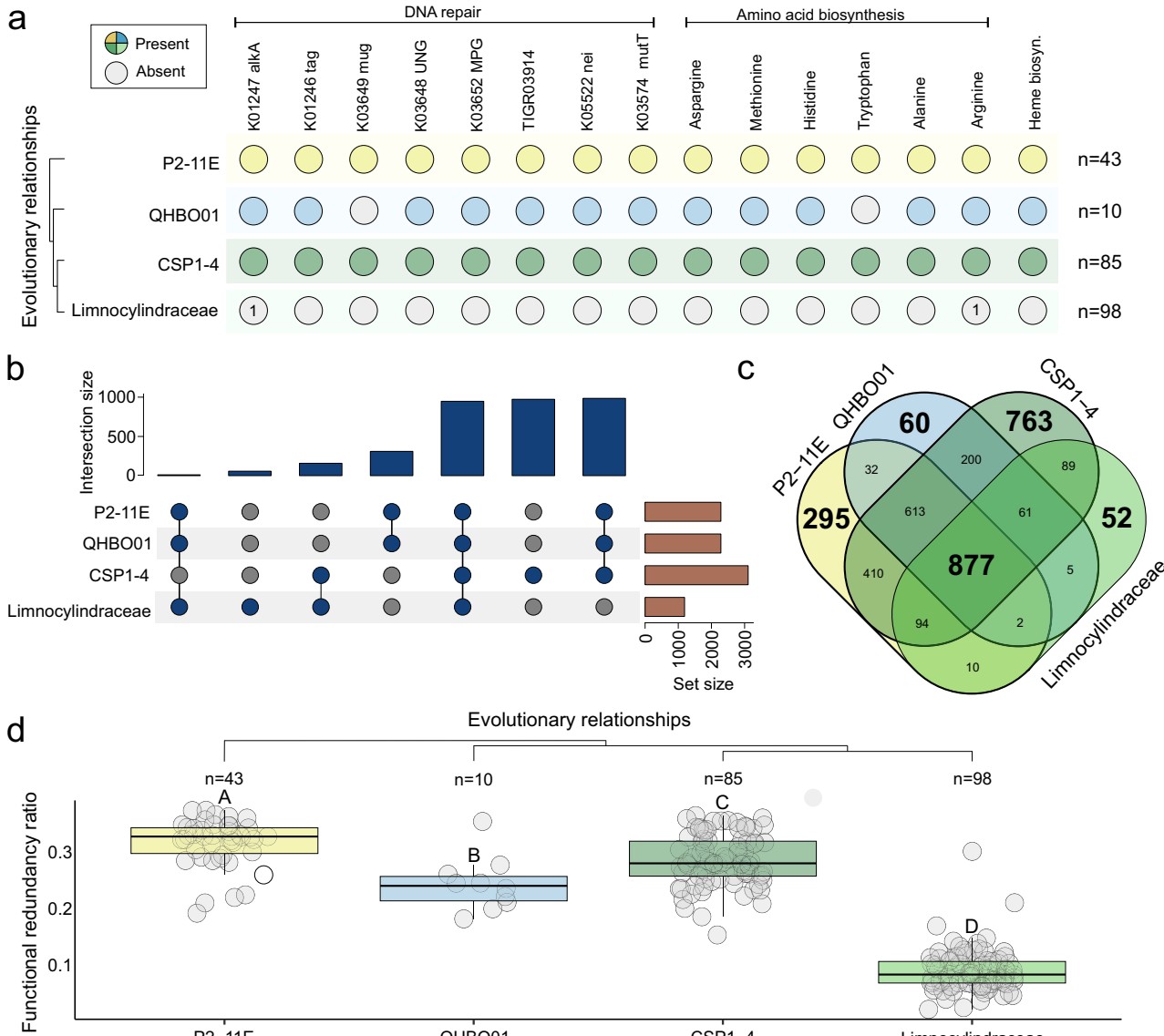

**Fig. 5 | Genome-informed metabolic reconstruction. a** Evolution of metabolic potential within Limnocylindria. The main evolutionary clades are displayed in a phylogenetic fashion, starting with the oldest branch. Colored circles indicate the presence of selected enzymes or metabolic pathways, while grey ones signify their absence. A value of '1' inside a grey circle denotes that the enzyme was detected in only one genome out of 98. The number of analysed genomes for each family is shown on the right side of the panel. **b**, **c** present the distribution of metabolic potential across the Limnocylindria class based on KEGG functional annotations. **d** Analysis of functional redundancy ratios, which measure the percentage of genes encoding redundant functions. A value closer to 0 indicates low occurrence of duplicated functional annotations, while a value near 1 suggests high functional redundancy. Pairwise statistical differences are highlighted via the means of capital letters. Boxplots sharing the same capital letter indicate no significant difference ($p > 0.05$), while boxplots with different capital letters are significantly different ($p \leq 0.05$). The central line across the boxplots identifies the median, marking the dataset's midpoint. The box itself demarcates the interquartile range, extending from the first quartile to the third quartile, encapsulating the central 50% of the data. The whiskers project from the box to the furthest data points not categorized as outliers and show the spread of the main body of the dataset. All statistics are reported in Supplementary Data S14. Source data are provided as a Source Data file.

transitions. Thus, the absence of DNA glycosylases may effectively explain the observed increase in mutation frequency, genome reduction, and decline in GC content in Limnocylindraceae.

**Proteome optimization**

Limnocylindraceae genomes are missing the core enzymes for six canonical essential-amino-acid pathways (Fig. 4a), indicating a substantial, though not complete, contraction of their biosynthetic repertoire. This partial auxotrophy likely heightens dependence on the external amino acids or peptide-rich organic matter. Freshwater lakes, which often contain dissolved amino acids and oligopeptides, provide a potential source of these essential substrates[49]. However, it remains unclear the extent to which reliance on external amino acids shaped the proteomic composition of Limnocylindraceae.

To investigate this, we calculated the C and N content of each protein and normalized these values by protein length (Fig. 6). Our analysis reveals that Limnocylindraceae possesses lower C and N contents per amino acid compared to its ancestors or the CSP1-4 sister lineage (Fig. 6a, b). When plotting the mean normalized values for C and N, we observe a direct relationship between the proteome's elemental requirements and a clear separation between Limnocylindraceae and the other clades, with the former exhibiting the lowest values (Fig. 6c). The loss of biosynthetic pathways for amino acids that are abundant in the proteome (e.g., alanine, arginine, histidine, etc.)

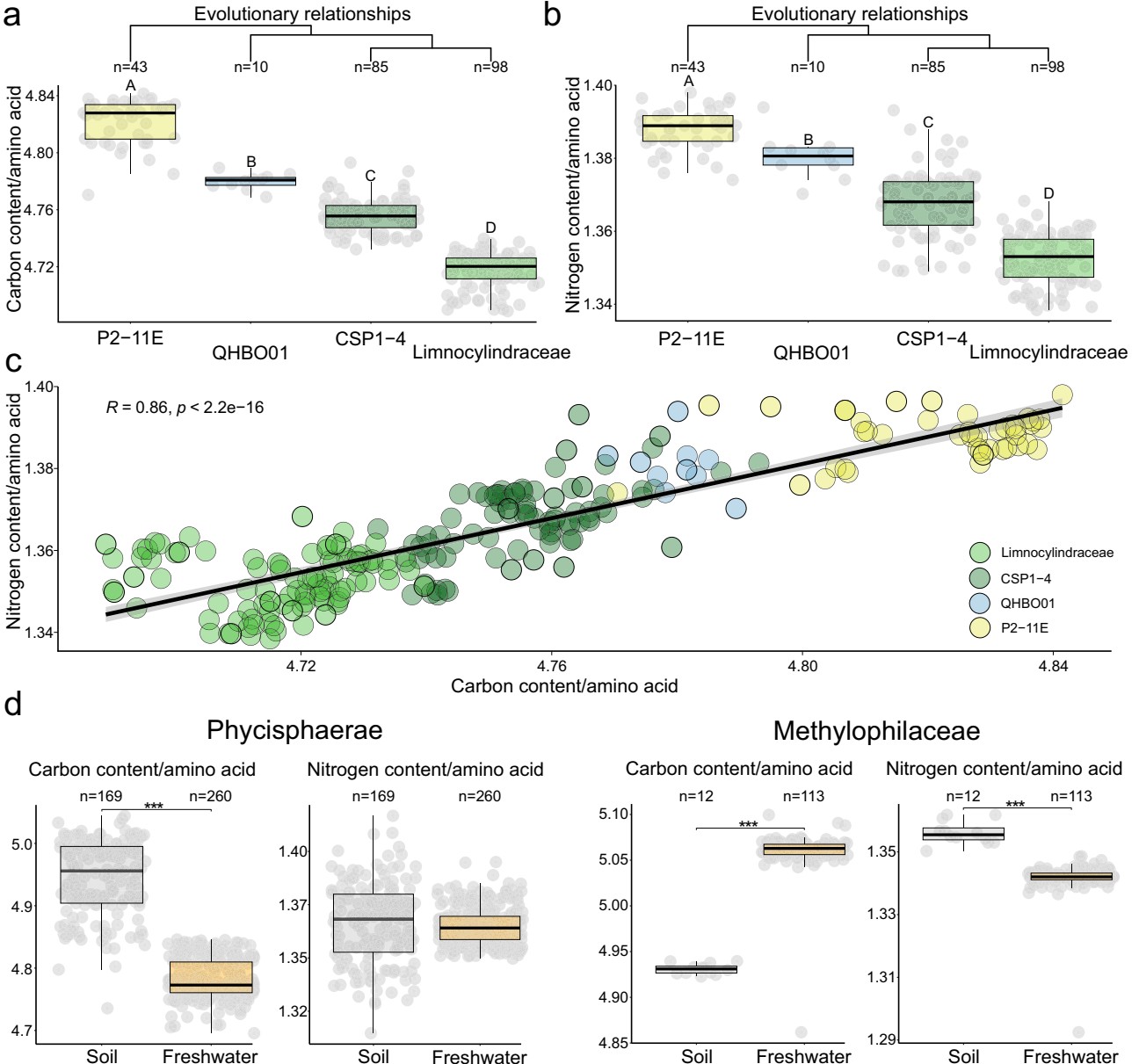

**Fig. 6 | Proteome content dynamics. a** Average amino acid carbon content. The *Y*-axis represents average carbon content per amino acid while the *X*-axis shows the taxonomic groups, arranged in evolutionary order. **b** Average nitrogen content as boxplots. The *Y*-axis depicts average nitrogen content per amino acid, while the *X*-axis shows the taxonomic groups ordered in a phylogenetic fashion. **c** Positive relationship between carbon and nitrogen content. The *Y*-axis shows the average nitrogen content per amino acid, while the X-axis displays the average carbon content per amino acid. Dots are colored according to the taxonomical labels, P2-11E (yellow), QHBO01 (blue), CSP1-4 (dark green) and Limnocylindraceae (green) (Spearman's rank correlation test: S = 303141, *p* value < 2.2e-16, rho = 0.8616216). The grey bands represent 95% confidence interval for the mean predicted by the regression. **d** Carbon and nitrogen content for Phycisphaerae (left) and Methylo-philaceae (right) lineages as boxplots. *X*-axis represents the average carbon and

nitrogen content per amino acid respectively, while the *Y*-axis shows the habitat provenance of the genomes. Soil/sediment lineages are represented as purple boxplots, while the freshwater ones are depicted as orange boxplots. Pairwise statistical differences are highlighted via the means of capital letters and stars (**p* < 0.05; ***p* < 0.01; ****p* < 0.001, ns ≤ 0.05). Boxplots sharing the same capital letter indicate no significant difference (*p* > 0.05), while boxplots with different capital letters are significantly different (*p* ≤ 0.05). The central line across the boxplots identifies the median, marking the dataset's midpoint. The box itself demarcates the interquartile range, extending from the first quartile to the third quartile, encapsulating the central 50% of the data. The whiskers project from the box to the furthest data points not categorized as outliers and show the spread of the main body of the dataset. All statistics are reported in Supplementary Data S14. Source data are provided as a Source Data file.

likely made it reliant on external environmental supply and exerted selective pressure to minimize the cellular demand for essential elements such as C and N. Thus, by optimizing its proteome to favor amino acids with lower C and N content, Limnocylindraceae likely reduce the metabolic cost associated with protein synthesis and maintenance.

To determine whether the proteome streamlining strategy is unique to Limnocylindraceae, we conduct two case studies focusing on the Planctomycetota (class Phycisphaerae) and Gammaproteo-bacteria (family Methylophilaceae) clades—groups that include linea-ges believed to have transitioned to freshwater environments[27,28]. Our analyses reveal that in both instances, the freshwater lineages

underwent C and N optimization, albeit in a niche-dependent manner (Fig. 6d). Specifically, freshwater Phycisphaerae exhibit overall lower C content per amino acid compared to their soil ancestors, reflecting a metabolism adapted for utilizing nitrogen-rich sinking aggregates[28]. Conversely, freshwater Methylophilaceae display reduced nitrogen requirements per amino acid, a trait that underlies their specialized metabolism for using reduced one-carbon compounds as energy and carbon sources[27]. These observations provide a novel perspective on previous studies suggesting that nutrient scarcity may foster proteome-wide nitrogen reduction, as demonstrated by comparative analyses of community-level protein sequences in open-ocean and coastal marine ecosystems[50]. Overall, our findings highlight the possible role of proteome streamlining as a broad evolutionary response to biosynthetic pathway loss and nutrient limitation following large habitat transitions.

## An eco-evolutionary synthesis

Bacterial ecosystem transitions have fundamentally shaped Earth's natural history by instigating transformative changes in the biosphere. For instance, the colonization of land by bacteria[51] is thought to have initiated biogeochemical rock weathering and soil formation, creating environments that later supported the emergence and diversification of early plants—a key milestone in life's evolution[52]. Yet, despite the recognized importance of such events, the selective pressures that drove them, as well as the evolutionary tempo (rates of change) and mode (patterns of adaptation), remain poorly charted.

The Chloroflexota phylum designates a major branch in the tree of life, thought to have emerged as a direct descendant of the earliest bacterial lineages that colonized land[51]. Among its clades, Limnocylindria is notable as one of the oldest and most prominent divisions[53]. This uncultivated lineage, identified through metagenomic surveys of freshwater lakes[54], derives its name from the aquatic habitats where its first representatives were recognized.

Our results indicate that within terrestrial Limnocylindria, the capacity for cross-ecosystem colonization likely emerges during a period of reduced selective pressure, as suggested by clade-specific selection force analyses (see Evolution of genomic architectures). In CSP1-4, cross-ecosystem colonization appears to have been driven by genome expansion and the acquisition of habitat-specific functions, whereas in Limnocylindraceae, the same ecological shift coincided with pronounced genome reduction—revealing two distinct evolutionary strategies within a shared evolutionary ancestry.

Ecological transitions in Limnocylindraceae and CSP1-4 exemplify niche-directed genome evolution[28], yet the sister clades appear to sit at opposite ends of that continuum. CSP1-4 presents genomic signatures consistent with diversifying selection accompanied by net gene acquisition. Soil representatives retain a broad plant-polymer catabolic arsenal; sediment-dwelling taxa carry additional nitrate-respiratory modules and microaerobic terminal oxidases; and freshwater representatives uniquely encode flagellar and chemotaxis loci—features uncommon in other Chloroflexota[34] (Supplementary Data S11). By contrast, Limnocylindraceae shows only modest habitat-linked gene gains, most notably the acquisition of proton-pumping rhodopsins (Supplementary Note 1).

This asymmetry may reflect divergent responses to relaxed selective constraint. In Limnocylindraceae, such relaxation appears to have permitted the accumulation of loss-of-function mutations in DNA repair genes—including Tag/AlkA/MPG, UDG-family, Nei, and MutT (Fig. 4a)—leading to the degradation of base excision repair (BER) pathways that defend against alkylated, deaminated, and oxidized bases. The resulting hypermutator state likely accelerated GC content erosion and contributed to the loss of multiple biosynthetic pathways (Fig. 5b). Thus, while CSP1-4 seems to have navigated habitat transitions through adaptive gene gain and genome expansion, Limnocylindraceae appears to have undergone reductive genome evolution

and metabolic simplification under sustained mutation pressure following BER collapse.

In Limnocylindraceae, the concerted loss of Tag-/AlkA-/MPG-mediated alkyl repair, UDG-driven uracil repair, Nei-directed oxidation repair, and MutT nucleotide sanitizing strips the lineage of its chief defences against the three most pervasive small-base lesions[42]. Because nucleotide-excision repair, mismatch repair, homologous recombination and AlkB-type dioxygenases cannot excise 3-methylguanine, uracil, thymine glycol or 8-oxo-dGTP with comparable efficiency, these lesions will likely persist into replication[42,45,55]. Laboratory strains deprived of the same activities manifest 10- to $10^4$-fold mutator phenotypes and strongly GC-eroding spectra—$G \rightarrow A$ (tag alkA), $G \cdot C \rightarrow A \cdot T$ (ung) and $GC \rightarrow TA$ (nei)[56–59]. Such biases furnish a direct mechanistic path to the lineage's AT-enrichment and gene loss, echoing broad phylogenomic evidence that links elevated mutation pressure to bacterial genome reduction[41] and the rapid GC erosion produced in BER-deficient *Salmonella typhimurium* evolution experiments[60]. Thus, in Limnocylindraceae, the capacity to cross ecological boundaries was likely accelerated by DNA glycosylase losses (Fig. 5a) that gave rise to hypermutator descendants (as inferred from branch length elongation, Fig. 1a) capable of rapid adaptation. Elevated mutation rates and ecological plasticity appear tightly linked; hypermutation likely enabled dramatic shifts in genome content and metabolic function, generating the genetic variation required to colonize novel environments. However, the very processes that facilitated this swift adaptation also sowed the seeds of constraint. The freshwater-adapted lineage evolved and diverged rapidly—as evidenced by its elongated phylogenetic branch, indicative of numerous non-synonymous substitutions—but eventually reached an evolutionary slowdown. Having shed a large part of its metabolic repertoire (Fig. 5b, c) and accumulated extensive sequence changes (Fig. 3), Limnocylindraceae ended up with a highly reduced genomic architecture. While this reduction likely facilitates niche specialization, the combination of genome shrinkage (Fig. 1b) and elevated mutation pressure (Fig. 3a) may also limit the lineage's capacity to explore new adaptive landscapes. Similar trade-offs have been documented in other microbial systems: in Prochlorococcus, genome reduction restricts physiological plasticity and niche range[61,62], while hypermutator strains of *Pseudomonas aeruginosa* lose fitness when introduced to new environments[63]. Thus, although Limnocylindraceae remains evolutionarily viable, its current genomic configuration may reduce future evolutionary flexibility—a possible example of the tension between hyperadaptability and hyperspecialization[63].

We observe that genome reduction precedes the decline in GC content (Fig. 1), indicating that deletion events outpace mutation accumulation during the transition to freshwater environments. Once Limnocylindraceae adapted to these ecosystems, negative (purifying) selection likely emerged as a dominant force (Fig. 3), preserving the primary structures of essential proteins. Concurrently, the loss of DNA glycosylases (Fig. 5a) most probably facilitated the accumulation of GC-lowering mutations[60], predominantly at third codon positions where they are often synonymous and minimally impact protein function. The pervasive negative selection across numerous genes (Fig. 3a) underscores that while mutations arose randomly, only those occurring at neutral sites persisted. We observe that the substitution of third codon positions with AT-rich nucleotides was a principal driver behind the decrease of GC content in freshwater-adapted Limnocylindraceae (Fig. 3, and Supplementary Fig. S9). Thus, it is probable that the intricate interplay between mutation-induced GC reduction and selective pressures preserving protein functionality orchestrates the retention of a high GC content in the genome-reduced freshwater lineage (Supplementary Fig. S10).

Genome size and GC content are hallmarks that define the architectural blueprint of bacterial genomes. Understanding the forces that shape these features has been a central pursuit of evolutionary

microbiology[23,32,64,65]. Traditionally, two paradigms—the adaptive[35] and the non-adaptive[66]—have competed to explain the remarkable variation observed across bacterial taxa. The adaptive paradigm portrays genomes as finely tuned instruments, honed by natural selection to maximize efficiency. Deterministic mechanisms like genome streamlining[35] act as evolutionary chisels, chipping away non-essential DNA to reduce genome size and minimize resources required for replication in response to nutrient limitations. The 'black queen hypothesis' enriches this narrative, suggesting that selective pressures eliminate costly communal functions, offloading them onto other community members and thereby lightening the genomic load[67]. Ancient adaptations are further integrated into this framework, with remnants of past selective pressures continuing to shape contemporary genomic traits such as GC content[68]. In stark contrast, the non-adaptive paradigm[69] posits that the evolution of genome size and GC content is more a product of chance than design. Here, neutral processes such as elevated mutation rates and random genetic drift—operating independently of selective pressures—introduce genomic variability[23].

Our findings in freshwater Limnocylindraceae support the non-adaptive paradigm, suggesting that its genome reduction and GC content are likely driven by mutational biases rather than selective optimization. Unlike other free-living marine bacteria that adaptively streamline their genomes and lower GC content in response to nutrient-poor environments[35], Limnocylindraceae genomic architecture is likely shaped by inherent processes linked to elevated mutation rates and the environmental niche it occupies. This rationale resonates with experimental evolution studies where GC content decrease in microbial isolates was attributed to mutational biases in the absence of DNA repair systems[60] and align with recent observations regarding genome reduction in Prochlorococcus[70].

Considering that DNA constitutes approximately 3% of a bacterial cell's dry weight while proteins account for about 53%[71], we hypothesized that any adaptive mechanisms in Limnocylindraceae would likely operate at the proteomic level. Our analysis of the carbon and nitrogen content of amino acids reveals that Limnocylindraceae preferentially utilize amino acids with lower C and N content (Fig. 6). This selective utilization could be particularly significant given that proteins represent the largest fraction of bacterial biomass. Thus, the transition to freshwater environments appears to have driven a strategic optimization of carbon and nitrogen usage within the proteome. This proteomic optimization likely emerged during the initial phases of adaptation to freshwater habitats, characterized by a substantial increase in non-synonymous substitutions (long branch in Fig. 1a). Furthermore, we show that similar proteome-level C or N optimizations occur in other microbial lineages believed to have transitioned into freshwater habitats (Fig. 6d), suggesting that proteome streamlining may represent a more widespread adaptive strategy among freshwater taxa.

Together, these findings reveal how divergent genomic strategies—one expansive and acquisitive, the other reductive and minimalist—can both drive microbial transitions across ecosystem boundaries. Arising from a shared terrestrial ancestor, CSP1-4 and Limnocylindraceae exemplify contrasting routes to ecological success: the former through gene acquisition and metabolic diversification, the latter through mutation-driven genome erosion and functional contraction. In Limnocylindraceae, genome reduction was highly likely catalyzed by the collapse of base excision repair, fostering a hypermutator state that promoted rapid genetic change and facilitated ecological flexibility. This mutational pressure reshaped genome architecture, eroded biosynthetic capacity, and constrained functional breadth. Yet even within this reductive trajectory, selective pressures at the proteomic level favored elemental efficiency, sustaining survival in nutrient-variable freshwater environments. These results demonstrate how cross-ecosystem transitions can reconfigure bacterial genomes through the intertwined forces of adaptive refinement and non-adaptive mutation dynamics, offering an expanded mechanistic framework for genome reduction and ecological specialization in free-living bacteria.

## Methods

### Limnocylindria genomic database

To build a comprehensive genomic database for the bacterial class-level lineage Limnocylindria, we began by reclassifying metagenome-assembled genomes (MAGs) from the recently published pdCEL v1.0 database[15] using GTDB-Tk[72] version 2.4.0 (database release R09-RS220). The pdCEL database contains approximately 5500 high-quality MAGs sourced from Central European freshwater lakes with varying trophic statuses. These MAGs were subjected to meticulous curation, including taxonomic validation, checks for GC content consistency, and the application of stringent thresholds for genome completeness and contamination to ensure data integrity[15]. From this reclassified dataset, we identified 72 Limnocylindria MAGs based on the updated taxonomy. To expand our dataset further, we incorporated an additional 164 MAGs indexed in the Genome Taxonomy Database (GTDB)[73], resulting in a curated collection of 236 high-quality MAGs. This extensive repository provides a robust foundation for in-depth genomic and evolutionary analyses, enabling us to investigate Limnocylindria adaptations across diverse environments such as soil, sediment, and freshwater.

To estimate genome sizes accurately, we corrected the assembled lengths of the MAGs for both completeness and contamination. This was achieved by dividing each MAG length by its estimated completeness and then multiplying the result by the difference between 100 and the MAG contamination value. This approach accounted for missing genomic content and excluded contamination[15].

Because short-read MAGs can under-recover parts of the flexible (accessory) genome, we applied conservative criteria for interpreting gene presence/absence. Specifically, we restricted downstream analyses to quality-filtered, curated MAGs and inferred lineage-level gene depletion only when a gene was consistently undetected across multiple MAGs, rather than from single-genome non-detections. Accordingly, "absence" is interpreted as loss from the conserved (core) gene repertoire at the lineage level, while rare strain-specific occurrences (e.g., via horizontal transfer) cannot be formally excluded.

### Evolutionary history reconstructions

Phylogenetic analyses were based on a set of 118 single-copy proteins previously identified as phylogenetically informative and suitable for deep evolutionary inference[74]. To maintain balanced representation and uphold the genomic diversity of Chloroflexota lineages, the initial collection of 236 genomes was subsampled to 50. This subsampling retained all 10 QHBO01 family representatives and chose additional genomes from other lineages to ensure their genomic diversity was adequately represented. Protein-coding genes were predicted using PRODIGAL[75] v2.6.3 with default settings for each selected MAG ($n = 50$). The resulting proteomes were scanned using HMMER hmmscan[76] v3.1b2 against the TIGRFAMs[77] v15.0 database (parameters: E-value ≤ 1E-10; -prcov 70; -hmcov 70). Based on TIGRFAM accession numbers, 118 ubiquitous single-copy protein sequences were extracted from the genomes. For each phylogenomic marker, phylogeny-aware multiple sequence alignments (MSAs) were constructed using PRANK[78] v.170427 with default settings. The alignments were trimmed with BMGE[79,80] v1.12 using the -g 0.5 setting to retain regions suitable for phylogenetic inference. The individual MSAs were concatenated into a supermatrix comprising 39,854 aligned sites prior to phylogenetic analysis. A maximum-likelihood phylogeny was constructed using IQ-TREE[80] v2.0.6. To account for amino acid frequency variability and evolutionary rate heterogeneity across sites, we employed mixture models with the LG substitution matrix and 30 composition profiles.

Node support was assessed using ultrafast bootstrapping with 1000 replicates and the Shimodaira-Hasegawa approximate likelihood ratio test (SH-aLRT) with 1000 replicates. Outgroup rooting was performed by using members of the Eremiobacterota phylum, chosen due to their phylogenetic position related to the Chloroflexota (1). This approach was performed to confer directionality to the phylogenomic tree. A jack-knifed family-centric genome-based phylogeny of the Limnocylindria class is further presented in Supplementary Fig. S11.

## Phylogenetic gain

To evaluate the increase in phylogenetic diversity from incorporating MAGs from the pdCEL database, we conducted phylogenetic analyses using both the newly added MAGs ($n = 72$) as well as the existing database representatives ($n = 113$) based on 118 single-copy proteins known to be phylogenetically informative and appropriate for deep evolutionary inference.

Protein-coding genes were predicted using PRODIGAL v2.6.3 with default settings for each selected genome ($n = 185$). The resulting proteomes were then scanned HMMER hmmscan v3.1b2 against the TIGRFAMs[77] v15.0 database (parameters: E-value ≤ 1E-7; -prcov 70; -hmcov 70). Based on TIGRFAM accession numbers, 118 ubiquitous single-copy protein sequences were identified and extracted from the genomes. For each phylogenomic marker, phylogeny-aware multiple sequence alignments (MSAs) were constructed using FAMSA v1.2[81] with default settings. All individual MSAs were concatenated into a supermatrix comprising 53,775 aligned sites prior to phylogenetic analysis. A maximum-likelihood phylogeny was constructed with IQ-TREE[75,80] v2.1.3, employing mixture models with the LG substitution matrix and 60 composition profiles. Node support was assessed using ultrafast bootstrapping with 1000 replicates and the Shimodaira-Hasegawa approximate likelihood ratio test (SH-aLRT) with 1000 replicates. Finally, the tree was rooted with two members of the P2-11E family ($n = 2$), chosen for their phylogenetic position relative to the newly added MAGs, thus providing directionality to the final phylogenomic tree (Supplementary Fig. S12).

Phylogenetic diversity (PD) was calculated as the sum of all branch distances in each subtree. The gain in PD introduced by the pdCEL MAGs was then assessed as follows: $PD_{total} - PD_{Reference\ genomes} = PD_{gain}$. All calculations were carried out within R v.4.2.2 with the ape v.5.3 package.

## Taxa-habitat associations

To assess the environmental distribution and relative abundances of Limnocylindria lineages, we utilized the Sandpiper v0.3.0 database[33]. Sandpiper integrates taxonomic profiles generated by the SingleM v0.20.3 software[33] from over 248,000 publicly available metagenomes. The main approach of SingleM is to profile metagenomes by targeting short 20-amino-acid stretches, or 'windows', within single-copy marker genes. It identifies reads that cover an entire window and analyzes these further. By focusing on these short windows, SingleM can determine how novel each read is compared to known genomes[33]. Leveraging the fact that each analyzed gene is typically found exactly once in each genome, the software accurately estimates the abundance of each lineage. We focused on data at the family level, selecting datasets where Limnocylindria lineages achieved abundances of at least 0.5% within the total prokaryotic community in each metagenome. By using this threshold, we recovered 14,644 valid taxa-habitat matches (i.e., one of the four family-level lineages achieved a total abundance in the prokaryotic community equal to or greater than 0.5%). This threshold minimized noise and ensured robust, ecologically meaningful associations between taxa and habitats.

## Protein annotations and functional redundancy analysis

Protein-coding genes were predicted, for each MAG ($n = 236$), using PRODIGAL[75] v2.6.3. Protein domains were annotated by querying the predicted proteomes against the Pfam[82] release 32 HMM database using the hmmscan-based pfam_scan.pl script. Additional domain architectures and protein annotations were obtained by running InterProScan[83] 13v5.24-63.0 with the CDD v3.14, SMART v7.1, and HAMAP v201701.18 databases. To further expand the protein annotation space, we conducted hmmsearch (parameters: -evalue 1E-7, -prcov 70, -hmcov 70) against the COG[84] and TIGRFAM[77] HMM databases. BlastKOALA v3.1[85] was utilized to assign KEGG Orthology (KO) identifiers to orthologous genes. Selected metabolic pathways were manually examined using the online KEGG mapping tools based on the assigned KO numbers. Rigorous manual curation was applied to ensure metabolic pathway completeness.

To assess habitat-specific patterns in functional gene content within the CSP1-4 lineage, we computed pairwise Jaccard similarity indices based on the presence or absence of KEGG Orthology (KO) identifiers. For each habitat (soil, sediment, and freshwater), KO terms detected across all associated CSP1-4 genomes were pooled into a composite gene set. The similarity between any two habitats $i$ and $j$ was calculated as:

$$J_{ij} = \frac{|K_i \cap K_j|}{|K_i \cup K_j|} \qquad (1)$$

where $|K_i \cap K_j|$ is the number of KOs shared by both habitats and $|K_i \cup K_j|$ is the total of unique KOs present in either habitat. This analysis enables a quantitative comparison of metabolic repertoire overlap across habitats and highlights functional divergence during ecological transitions within CSP1-4.

To assess functional redundancy in the proteins of selected bacterial species, we calculated the Functional Redundancy Ratio (FRR)[15]. For each MAG, we determined the total number of KEGG annotations from all the protein-coding genes and the number of unique KEGG annotations representing non-redundant functions. The FRR was computed as one minus the ratio of unique to total KEGG annotations per MAG. An FRR value near 0 indicates low functional redundancy, while a value near 1 indicates high functional redundancy within the genome.

## Ancestral state reconstructions

To resolve the evolutionary trajectories underpinning ecological transitions and metabolic adaptation in Chloroflexota, we performed ancestral state reconstructions on a maximum-likelihood phylogeny comprising 50 representative genomes (outgroups excluded).

Habitat transitions were inferred by modeling the evolution of discrete ecological states—soil, sediment, and freshwater—using stochastic character mapping as implemented in the make.simmap() function (phytools v2.0, R). Three standard Markov (Mk) models were evaluated: the Equal Rates (ER) model, in which all transitions occur at the same rate; the All-Rates-Different (ARD) model, allowing unique rates for each transition; and the Symmetric (SYM) model, where forward and reverse transitions share identical rates. Model selection was based on log-likelihood scores and Akaike Information Criterion (AIC) values. The ER model provided the best fit and was subsequently used to generate 1000 stochastic character maps, yielding posterior probabilities of ancestral habitat states at each node.

Trait evolution was reconstructed for 15 binary-coded pathways related to sensory perception and core metabolism, including rhodopsin, flagellar motility, chemotaxis, heme biosynthesis, and nine amino acid biosynthetic pathways (e.g., arginine, methionine, tryptophan). Each trait was modeled independently under an ER framework using fitMk() and stochastically mapped with make.simmap() (1000 replicates per trait). Posterior node probabilities were summarized via describe.simmap() to infer the timing and directionality of trait gain or loss across major habitat transitions.

These analyses enabled the reconstruction of metabolic and sensory innovations (or regressions) associated with lineage-specific ecological specializations and are summarized in Supplementary Fig. S14.

### Selection force

Selection force analyses were conducted as previously described[15]. Briefly, gene sequences from functionally annotated protein orthologous groups—870 for Limnocylindria family-level analyses and 594 for Limnocylindrus species-level analyses—were extracted in FASTA format. To prepare the nucleotide sequence data for downstream analyses, we used a custom bash script designed to remove terminal stop codons from coding DNA sequences. For each sequence, the script checked the last three nucleotides to determine if they correspond to one of the standard stop codons (TAA, TAG, or TGA). If a terminal stop codon was detected, the script removed it from the end of the sequence. Subsequently, the preprocessed sequences were aligned using PRANK[78] v.170427 in codon mode (-codon -F). Alignments containing three or more sequences were then used to construct phylogenetic trees with IQ-TREE[80] v2.1.3, utilizing the codon model setting (-st CODON). Site-specific selection pressures were inferred from the codon alignments and phylogenetic trees using the Fixed Effects Likelihood[86] (FEL) method implemented in HYPHY v2.5.32 within a maximum-likelihood framework, applying a p-value threshold of 0.05.

Genes from functionally annotated protein orthologous groups from the four Limnocylindrus (*L. sp003670745*; *L. sp903937225*; *L. sp003670475*; *L. sp903864185*) species were extracted in FASTA format. Orthologous groups shared among all four species were discarded, leaving only those that were unique to each species (*L. sp003670745 n = 26*; *L. sp903937225 n = 37*; *L. sp003670475 n = 23*; *L. sp903864185 n = 69*). GC content for each gene was then calculated using seqstat v1.9 g (--gccomp -a). The same pipeline was run on the shared orthologous groups ($n = 687$) by all four *Limnocylindrus* species to provide a baseline for comparison.

### Codon and amino acid frequency analyses

Codon frequencies were calculated using a custom Perl-based script that reads coding DNA sequences from multi-FASTA files and counts the occurrences of all 64 possible codons by iterating through the sequences in frames of three nucleotides. The total counts for each codon are then normalized by dividing by the total number of codons in each genome, resulting in codon frequency values. The script outputs a tab-delimited file where each row corresponds to a codon and each column represents the codon frequencies for a particular genome. A total of 236 genomes were analysed.

Amino acid compositions of predicted protein sequences were calculated using computational methods employing the BioPerl[87] module Bio::SeqIO. Protein sequences in FASTA format were read and processed individually. For each protein sequence, the occurrences of all 20 standard amino acids were counted. These counts were then normalized to percentages relative to the total sequence length, providing the amino acid composition profiles used in our analyses.

### Predicted proteome carbon and nitrogen content

The carbon and nitrogen content of proteins ($n = 462,342$) encoded in the genomes belonging to the studied Chloroflexota families (i.e., P2-11E, QHBO01, CSP1-4, and Limnocylindraceae) was determined by using a custom BASH script. Briefly, the script counted the occurrences of each amino acid and computed the carbon and nitrogen content using predefined molecular composition values. The carbon and nitrogen content of each protein was normalized by the corresponding predicted protein length, then averaged to obtain the mean carbon and nitrogen content per proteome.

To complement the analysis, the dataset was expanded to include additional lineages hypothesized to have transitioned to freshwater environments, respectively Planctomycetota (class Phycisphaerae[28]) and Gammaproteobacteria (family Methylophilaceae[27]) clades (3). For each lineage, freshwater (Phycisphaerae $n = 260$; Methylophilaceae $n = 113$) and soil-sediment (Phycisphaerae $n = 169$; Methylophilaceae $n = 12$) representatives were recovered from GTDB[73] (release R09-RS220). Sandpiper[33] database was then used to further validate their habitat distribution.

### CARD-FISH

For the CARD-FISH analysis, we collected freshwater samples (10 mL) from the epilimnion (5 m depth) of Lake Zurich (406 m a.s.l., 47°18′N, 8°34′E, Switzerland). The samples were immediately fixed with formaldehyde at a final concentration of 2% and incubated at room temperature (25 °C) for 1 h. After fixation, the samples were filtered through 0.2 μm polycarbonate membrane filters (Merck Millipore, Germany) supported by 5 μm cellulose nitrate filters (Sartorius, Germany), followed by double washing with phosphate-buffered saline (PBS, 1X). The filters were air-dried and stored at -20 °C until further use.

CARD-FISH was performed using a probe specific for the Limnocilindraceae lineage[54] of Chloroflexota bacteria. Probes NON338 (negative control) and EUB I-III[88] (positive control) were included to validate the results. All filters were in addition stained with DAPI (4′,6′-diamidino-2-phenylindole) and examined using an Axio Imager.M1 microscope (Zeiss, Germany). Micrographs of stained cells were captured with a Nikon DS-Qi2 Monochrome Microscope Camera (Nikon, Japan) and processed using NIS-Elements Advanced Research software v5.01.00.

### Statistics and code

All statistics were performed within the R[89] v4.2.2 (R Core Team, 2022) and RStudio[90] v1.3.1093 (RStudio Team, 2020), Apricot Nasturtium software. Shapiro-Wilk test, followed by residues distribution visualization, was utilized for data normality assessment. Parametric datasets analysis of variance (ANOVA) was performed for normally distributed datasets. Pairwise comparisons at group level were performed with pairwise t.test whenever the dataset was normally distributed. Non-parametric datasets analysis of variance was performed with the Kruskal-Wallis tests. Subsequently, pairwise Wilcoxon rank sum test was performed for pairwise comparisons. For non-normal samples, variable relationship assessment was done by using the Pearson-rank correlation test. Discussed tests were performed using the corresponding functions within the stats v.4.2.2 package. The stats package is part of R. To assess whether the mean values differed significantly between groups that were unbalanced in both size and variance, a permutation-based t test was performed (999 permutations) using the t.perm.R function.

No statistical method was used to predetermine sample size. No data were excluded from the analyses. The experiments were not randomized. The investigators were not blinded to allocation during experiments and outcome assessment.

### Reporting summary

Further information on research design is available in the Nature Portfolio Reporting Summary linked to this article.

## Data availability

All genomic data utilized in this study is publicly available. 236 MAG IDs and their NCBI accession numbers are provided in Supp. Data S1. NCBI SRA IDs are provided for 14,644 metagenomes in Supp. Data S5. All additional important data supporting the study's conclusions are included in this publication and its Supp. Data 2–15. Source data are provided with this paper.

## Code availability

The source code generated in this study is publicly available on GitHub at (https://github.com/MiELevog/Cross-ecosystem-colonization) (https://doi.org/10.5281/zenodo.18416940)[91].

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

## Acknowledgements
We are grateful to Jakob Pernthaler for his thorough and constructive review of the manuscript draft. A.-S.A. and L.S.M. were supported by the Ambizione grant PZ00P3_193240 (Swiss National Science Foundation). C.H. was supported by the research grant 10000877 (Swiss National Science Foundation).

## Author contributions
Conceptualization: A.-S.A. and L.S.M. Bioinformatics: L.S.M., C.H., and A.-S.A. CARD-FISH: A.S. Statistics: L.S.M. Writing, original draft: A.-S.A. with input from L.S.M. Writing, review, and editing: L.S.M., A.S., and C.H. Funding acquisition: A.-S.A.

## Competing interests
The authors declare no competing interests.
