## [Transparent Peer Review file · Nature Communications]

Deep-branching Chloroflexota lineages illuminate the eco-evolutionary foundation of cross-ecosystem colonization

Corresponding Author: Dr Adrian-Stefan Andrei

Version 0:

Reviewer comments:

Reviewer #1

(Remarks to the Author)

OVERVIEW: This paper explores the genomic and phylogenetic properties of a set of related Chloroflexi lineages spanning multiple ecosystem types. The authors establish a robust phylogenetic framework for these lineages and undertake a series of analyses of genomic characteristics, genomic architecture, and to some extent, metabolic capacities to ostensibly illuminate the processes governing their transitions across ecological boundaries. While I commend the authors on an intriguing paper, I find numerous major conceptual issues that should be addressed before publication.

The main issue I perceive, as detailed in the below MAJOR COMMENTS, is that the paper relies far too heavily on speculative statements about essentially unknowable past evolutionary events. Throughout the text, the authors confidently assert a particular evolutionary path for their lineage of interest, which I do not believe to be fully substantiated by the evidence presented. Relatedly, the authors fail to explore alternative explanations for the ecological patterns they observe, seemingly relying heavily on a favored hypothesis about hypermutation as a result of the loss of one particular set of DNA repair genes. However, alternative repair systems, ecological pressures, or metabolic changes are not explored. To improve the robustness of the work, I recommend drastically revising the text to reduce causal/speculative language (focusing more on traits associated with habitat transition, rather than those causing it) and exploring alternative explanations for the intriguing correlations observed.

Unfortunately, this may also mean less solid ground to stand on in terms of 'reconcil[ing] adaptive and non-adaptive paradigms' applicable more generally to Bacteria. While I appreciate the effort to assess other lineages, the generalizability of the study would be increased if this section was further fleshed out.

MAJOR COMMENTS:

l143 - do sediment-associated lineages within CSP1-4 represent coherent phylogenetically coherent clade(s)? while soil and sediment habitats clearly have structural similarities, the transition to (aquatic?) sediments may still entail genomic/metabolic change (e.g. acclimation to low oxygen, depending on the aquatic system/sediment stratum). this habitat transition is currently not really explored in the text but doing so would give us a broader sense for the processes at play within this lineage.

fig 1 - in general, i think adding habitat affiliation to the tree on a per-species basis, if possible, would help the reader to better grasp where habitat transition events are likely to have occurred over evolutionary time (for both Limno and CSP1-4).

line 184 - when making this claim it is important to qualify with the degree of phylogenetic novelty accounted for by the new genomes. how many distinct species, genera, etc.. are included within this 68%? A large number of genomes from 1 or 2 heavily sampled species might be less significant than a broader sampling effort in this case.

line 207-209 - while i am (tentatively) on board with the likening of ancestral genomic features based on short branch lengths within a ribosomal tree (though i think it would be safer to just stick to overall 'divergence', rather than speculating about specific features), inferring that modern lineages experience similar eco-evolutionary pressures to more ancient ones is too speculative, at least for the results section.

line 223 - what is meant by 'align'? I think one could safely say correlate...

line 226 - again, it is unclear to me how methods based on modern genome sequences allow you to speak to selection pressures of the past.

line 246 - is it possible that the acquired genes (or a subset of them) themselves explain the changes in GC? or is there evidence for GC remodeling more generally across the genome? this seems important to rule out among otherwise closely related species.

line 298 - is there any precedent for these curve shapes/plots in the literature? I do not think there really is a justification for connecting four discrete points with a trend line. if you must, then please visually indicate the points (is it just one per lineage per codon?)

line 386 - while the hypothesis advanced here is interesting, it feels incomplete. are there other repair systems that could compensate for the lack of dna glycosylases? it would seem important to rule this out before making the claim that this lineage is uniquely mutation-prone, and even further, that this mechanism may underlie genome-wide GC shifts, especially without supporting experimental evidence.

line 405-408. What is the purpose of the DAPI stained image? Or of the part of the graph that shows presence/absence of bacteriorhodopsin? As far as I can tell, neither of these are referenced in the main text and thus are confusing (also panels are mislettered). please either address in the main text or delete.

410-413. Unfortunately, this sort of speculation has no place in a results section. what supports your claim that elevated mutagenesis led to the loss of AA biosynthetic pathways? from my perspective, the development of metabolic interdependencies/interactions could just as easily explain this observation (see the CPR bacteria as an example). I would strongly consider moving these sentences to the discussion section and motivating this analysis differently.

497-99, 532-34. While an interesting hypothesis, again, I do not think you can confidently conclude this evolutionary trajectory based on the evidence presented. The authors seem to focus on a favored hypothesis here (loss of DNA glycosylase-> hypermutation/plasticity) but do not explore alternative hypotheses. what about the rest of the genome? What other metabolic differences are observed between the freshwater and soil/sediment clades? As the authors themselves note in the introduction, these changes too may help explain the ability of lineages to cross ecological boundaries. Without exploring these alternatives fully, the conclusion risks coming across as cherry-picked.

510-516. Again, what evidence substantiates these claims about evolutionary dead ends? If there is literature you can cite in service of this argument, please do so, otherwise this is entirely speculative and should be written less assertively.

MINOR COMMENTS:

I12 - maybe replace 'unparalleled' here. while of course dependent on bacteria, viruses are also present in nearly every ecosystem on earth as well.

I17 - what is meant by 'integrative'? perhaps clearer to phrase as 'integrating X and Y'...

I133 - nice use of sandpiper here!

I194 - why are sigma factors being singled out here, among all genomic features?

Reviewer #2

(Remarks to the Author)

This manuscript investigated the genomes of Limnocyndria, a deep branching clade of the Chloroflexota phylum. The authors aimed to illuminate the mechanisms underpinning microbial evolution during major ecosystem transitions and foundations of bacterial adaptability. They used a database of genomes to determine the directionality of ecosystem colonization events by this lineage, and report that during the habitat transition from soil to sediment to freshwater, non-adaptive processes shaped genome size and GC content. Additionally, the purging of the DNA glycosylase gene pool allowed for flexible genomic architecture. Meanwhile, the adaptive mechanism of proteome streamlining supported the survival of the lineage in the new ecosystem(s).

The research represents important evolutionary work addressing how microbial species are able to transition from one environment to others, a knowledge gap with few well-demonstrated examples in the literature. Chloroflexota is a good lineage with adapted lineages to a variety of environments, making it ideal to study to address associated questions. Overall, the methods are generally sound and the paper is well written.

However, the manuscript does at times infer and speculate on mechanisms that drive the adaptability and survival of the lineage during ecosystem transitions, particularly in the results section. Specific examples are noted in the line specific comments, and I would recommend a revision to address some of the perhaps overly speculative correlations. Additionally, it may be helpful to incorporate the results of additional methods to uncover ancestral genes using ancestral state reconstructions. I would recommend either a rephrasing of some inferences in the results/discussion, or the demonstration of

these processes with additional methods.

Specific comments:

Throughout the manuscript, the terms non-adaptive vs adaptive selection is used. However, negative vs. positive selection, and purifying vs. diversifying selection are also terms that are used. Please forgive me if there was a purpose for all these different terms in particular usages, but it would help a reader if one set of terms was consistent throughout the manuscript.

Line 26: Related to Figure 2D - the GC content of the freshwater adapted clade is lower than its ancestors, is this truly "exceptionally high"?

Line 68-82: What about Actinobacteria? See work such as Ghai et al., 2013 (Metagenomics uncovers a new group of low GC and ultra-small marine Actinobacteria)

Line 129-132: Was this done with an outgroup?

Line 152: Figure 1 - the green colors transitioning from A to B are kind of confusing given the two sister clades are red and green in A, but only green in B. Maybe indicate both clades as green in A?

Line 201-209: This section in particular seems speculative - could be tested with ancestral state reconstruction.

Line 242: What is species defined as? (ANI/AAI?)

Line 248: No changes in what?

Line 251: Not able to tell what are synonymous substitutions from Figure 2, unless all of the data depicted in Figure 2 is from dS? If so, this needs to be stated in the caption.

Line 257: Figure 2D - could these differences be due to noise (not true signal) due to undersampling? Also, the GC content of the freshwater adapted clade is still lower than its ancestors (see line comment on the abstract).

Line 282-285: This seems speculative - is higher GC content a consequence of small size? What is the mechanism there?

Line 285: Consider discussing implications for "Increased mutation rate is linked to genome reduction in Prokaryotes" (Bourguignon et al., 2020) within this sentence.

Line 288-289: Again no mechanistic explanation here.

Line 290-308: Ok I see the reasoning behind some of the mechanism for the GC content changes driving the cell shape, but the size link is still unclear in the previous paragraph.

Line 389: Figure 4 - consider matching colors with figure 1.

Line 442-449, and throughout: This section in particular seems like it belongs in a discussion section, not results (brings in previous studies, and has a defined conclusion sentence) - consider revision of the text to separate results from discussion, or to have one combined results and discussion section. This applies to other sections throughout the results.

Line 450: Figure 5 - again consider matching colors with figure 1 and 4, and others.

Version 1:

Reviewer comments:

Reviewer #1

(Remarks to the Author)

I commend the authors for comprehensively and thoughtfully addressing my last round of comments. Their changes to language and addition of several supporting analyses have strengthened the manuscript.

At this point, I have only two additional comments:

1) The abstract is still too strongly worded for what is essentially an evolutionary hypothesis that cannot / is not proven definitively in the text. I suggest phrasing as 'we postulate' or 'we speculate' that.. based on X or Y evidence. As written, it is still far too deterministic.

2) Please incorporate citations to recent reviews covering microbial habitat transitions, including from/to freshwater environments. You may find these interesting points of comparison for your own work:

<https://doi.org/10.1111/1462-2920.16313>

<https://www.annualreviews.org/content/journals/10.1146/annurev-micro-041320-032304>

Reviewer #3

(Remarks to the Author)

I am reviewing a revised version of the manuscript. Overall, the manuscript reads well, and I consider that the authors have satisfactorily addressed most of Reviewer-2's comments.

One point I think should be discussed further is whether the quality of the MAGs could have influenced estimates of gene loss or genome reduction. As short-read (Illumina) MAGs typically miss genes from the accessory genome, I wonder whether the absence of genes (e.g., DNA glycosylases) could have been overestimated in specific lineages. In addition, were these MAGs manually curated to ensure that all contigs belong to the MAGs?

Point-by-Point Response to Reviewers

Manuscript title: "***Deep-branching Chloroflexota lineages illuminate the eco-evolutionary foundation of cross-ecosystem colonization***"

Dear Reviewers,

We are sincerely grateful for the time, care, and expertise you invested in reviewing our manuscript. Your detailed and constructive comments have been invaluable in helping us **reassess our analyses, clarify our exposition**, and present our findings with **greater precision**. We recognise that several of the concerns you raised stemmed from shortcomings in how we originally presented certain data and framed key arguments. In response, we have comprehensively reorganised the figures, enriched the methodological descriptions, and tempered speculative language so that our conclusions are now articulated with full transparency and evidential support. We trust that these revisions address your points in full and materially strengthen the manuscript.

In response to the comments received, we implemented the following major changes:

1. Enhanced clarity and tone - We thoroughly revised the manuscript for improved readability and precision, softening speculative language and ensuring claims are consistently supported by data or literature.
2. Refined focus - Following reviewer suggestions, we increased emphasis on the *CSP1-4* clade, providing a clearer parallel with *Limnocylintranceae* and better contextualizing their contrasting evolutionary trajectories.
3. Metabolic resolution - We expanded our metabolic analyses, including new reconstructions and habitat-specific comparisons, particularly emphasizing the ecological and genomic distinctiveness of *CSP1-4* and *Limnocylintranceae*.
4. Ancestral state reconstruction - We incorporated new ancestral state reconstructions for both habitat and metabolic traits, adding mechanistic support for our hypotheses on gene gain, loss, and adaptation across transitions.
5. Strengthened mechanistic inference - We added literature-supported biochemical evidence (e.g., from DNA glycosylase loss and hypermutator phenotypes) to bolster our proposed model of mutation-driven genome reduction in *Limnocylintranceae*.

Below, we respond to each of your comments in sequence, explaining how your feedback has been incorporated and where additional analyses or clarifications have been added. We quote the reviewers' remarks in italics and present our replies in regular text (rendered in dark blue). **All line numbers cited correspond to the revised manuscript version with tracked changes.**

REVIEWER 1

Comment 1: *“OVERVIEW: This paper explores the genomic and phylogenetic properties of a set of related Chloroflexi lineages spanning multiple ecosystem types. The authors establish a robust phylogenetic framework for these lineages and undertake a series of analyses of genomic characteristics, genomic architecture, and to some extent, metabolic capacities to ostensibly illuminate the processes governing their transitions across ecological boundaries. While I commend the authors on an intriguing paper, I find numerous major conceptual issues that should be addressed before publication.*

The main issue I perceive, as detailed in the below MAJOR COMMENTS, is that the paper relies far too heavily on speculative statements about essentially unknowable past evolutionary events. Throughout the text, the authors confidently assert a particular evolutionary path for their lineage of interest, which I do not believe to be fully substantiated by the evidence presented. Relatedly, the authors fail to explore alternative explanations for the ecological patterns they observe, seemingly relying heavily on a favoured hypothesis about hypermutation as a result of the loss of one particular set of DNA repair genes. However, alternative repair systems, ecological pressures, or metabolic changes are not explored. To improve the robustness of the work, I recommend drastically revising the text to reduce causal/speculative language (focusing more on traits associated with habitat transition, rather than those causing it) and exploring alternative explanations for the intriguing correlations observed.

Unfortunately, this may also mean less solid ground to stand on in terms of ‘reconcil[ing] adaptive and non-adaptive paradigms’ applicable more generally to Bacteria. While I appreciate the effort to assess other lineages, the generalizability of the study would be increased if this section was further fleshed out.”

Response 1: We sincerely thank Reviewer 1 for this comprehensive overview. Your comments have been instrumental in refining both the presentation and the clarity of our study. In response, we have undertaken four overarching improvements:

- 1) **Calibrated interpretation.** We have replaced definitive assertions with conditional wording—explicitly stating assumptions and using qualifiers (e.g., “likely,” “suggests,” “plausible”) wherever inferences derive from indirect evidence.
- 2) **Robustness evaluations.** We confirmed that our principal findings remain consistent when phylogenies are inferred under alternative substitution models and when analyses are repeated on jack-knifed genome subsets (see Suppl. Fig. S11).
- 3) **Ancestral-state reconstructions.** We performed stochastic character mapping of habitat affiliations and metabolic traits to test alternative evolutionary scenarios and reinforce statistical support for the soil → sediment → freshwater transition.
- 4) **Trait-focused narrative.** Following your suggestion, we now emphasize observable phenotypic and ecological traits over speculative causal mechanisms and have expanded the comparative analysis of the *CSP1-4* sister clade to contextualize transitional patterns.

We will address each specific point you raised in turn below, citing revised line numbers and figures to illustrate the exact changes. We hope these overarching revisions, together with the detailed, point-by-point responses that follow, fully address your concerns.

Comment 2: "l143 - do sediment-associated lineages within CSP1-4 represent coherent phylogenetically coherent clade(s)? while soil and sediment habitats clearly have structural similarities, the transition to (aquatic?) sediments may still entail genomic/metabolic change (e.g. acclimation to low oxygen, depending on the aquatic system/sediment stratum). this habitat transition is currently not really explored in the text but doing so would give us a broader sense for the processes at play within this lineage."

Response 2: We appreciate the reviewer's thoughtful feedback regarding the CSP1-4 lineage. We concur that habitat transitions, particularly from soil to sediment, are likely accompanied by genomic and metabolic adaptations within CSP1-4 lineages. Although information pertaining to taxonomy and habitat specificity was previously presented in the Supplementary Information (Supp. Data S1), we acknowledge that these findings were not explicitly discussed in the main text.

In response to the reviewer's suggestion, we conducted additional metabolic and genomic analyses specifically targeting CSP1-4 members. These new analyses reveal that soil-, sediment-, and freshwater-associated species form phylogenetically coherent subclades (i.e., species), each characterized by habitat-dependent genomic and metabolic adaptations. To more prominently present these findings, we have substantially expanded the main text (lines 241-266), introduced a new figure (Fig. 2), and added a new supplementary information file (Supp. Data S12). The following text has been incorporated into the manuscript:

"The CSP1-4 family comprises 16 genera and 51 strictly habitat-specific species—35 from soil, 4 from sediment, and 11 from freshwater. Genome-scale metabolic reconstructions, evaluated via a pairwise similarity matrix, revealed the greatest metabolic overlap between soil and sediment lineages (Fig. 2a,b). Soil genomes uniquely encoded enzymes for depolymerising plant polysaccharides and polyols (e.g., sucrose phosphorylase, 6-phospho- β -glucosidase, inosose dehydratase), along with nutrient-stress regulators (cstA, phoH/phoL, and phoB), reflecting adaptation to complex carbon inputs and episodic carbon-phosphate scarcity (Supp. Data S12). In contrast, sediment lineages harboured the catalytic α - and β -subunits of the nitrate reductase/nitrite oxidoreductase complex, a complete nitrate/nitrite ABC transporter, and the high-affinity cytochrome d terminal oxidase, consistent with respiration in oxygen-poor, nitrate-rich sediments. Freshwater representatives were distinguished by expanded flagellar and chemotaxis loci—traits rare in Chloroflexota—suggesting motility-driven navigation of heterogeneous aquatic niches (Supp. Data S12). Genomic architecture differed consistently among habitats, reflecting distinct ecological constraints (Fig. 2c-f). Freshwater genomes had the largest estimated sizes and encoded markedly fewer sigma factors than those from soil or sediment, indicative of regulatory streamlining in planktonic environments. GC content and coding density were elevated in both sediment and freshwater lineages relative to soil representatives, suggesting habitat-specific selection for more compact and information-dense genomes outside terrestrial ecosystems. Together, these habitat-specific patterns in gene content and genome structure support distinct ecological strategies shaped by carbon availability, nutrient dynamics, and redox conditions across terrestrial and aquatic environments (further details on Limnocyndria metabolism are provided in the Supplementary Discussion)."

Comment 3: "fig 1 - in general, i think adding habitat affiliation to the tree on a per-species basis, if possible, would help the reader to better grasp where habitat transition events are likely to have occurred over evolutionary time (for both Limno and CSP1-4)."

Response 3: We thank the reviewer for this valuable suggestion. In response, we have amended Fig. 1 to incorporate per-species habitat annotations and expanded our analytical framework to infer ancestral habitat transitions:

A. Tip-level habitat mapping

- Each of the 50 terminal nodes is now marked with a habitat-specific color: grey (soil), purple (sediment), or orange (freshwater), making extant ecosystem affiliations immediately visible.

B. Habitat ancestral-state reconstruction

- We employed the `fit.mk()` function in `phytools` v2.0 package to fit three discrete-state Markov models—Equal-Rates (ER), All-Rates-Different (ARD), and Symmetric (SYM)—to our maximum-likelihood tree (outgroup excluded).
- Model selection based on log-likelihood and Δ AIC identified the ER model as the best-supported.
- Under the ER framework, we performed 1,000 stochastic mappings (with the `make.simmap()` function) to estimate posterior probabilities of soil, sediment, and freshwater states at each internal node.
- Nodes in Fig. 1 are now additionally color-graded according to their mean posterior probabilities, providing a visual representation of ancestral habitat states.

C. Materials and Methods

- We have added a new subsection, "Ancestral State Reconstruction," detailing habitat-assignment criteria, model fitting, model selection procedures, and the stochastic mapping approach.

D. We expanded the Fig. 1 legend (lines 200-204) to include the following: "Node insets represent pie charts. At each internal node, the inset pie chart shows the marginal posterior probability of each ancestral habitat occupancy (inferred under the Equal-Rates model with 1,000 stochastic mappings); slice areas are proportional to these probabilities, and colours correspond to habitat categories. Tip labels are coloured according to the habitat from which each genome was recovered."

We believe that these enhancements substantially improve the manuscript by enabling readers to trace habitat-transition events both visually and quantitatively, thereby providing a deeper understanding of the evolutionary dynamics of environmental specialization within *Limnocyndria*.

Comment 4: "line 184 - when making this claim it is important to qualify with the degree of phylogenetic novelty accounted for by the new genomes. how many distinct species, genera, etc.. are included within this 68%? A large number of genomes from 1 or 2 heavily sampled species might be less significant than a broader sampling effort in this case."

Response 4: We thank the reviewer for emphasizing the importance of qualifying phylogenetic novelty beyond the number of genomes alone. Although these data were previously included in Supp. Data S1, we agree they should be clearly stated in the main

text. In response, we have now specified both (i) the number of newly sampled species (five) and (ii) the corresponding gain in phylogenetic diversity (Faith's PD = 48.14%).

The revised sentence (lines 230-233) now reads: "...and *Limnocylintranceae* ($n = 67$ genomes, representing five novel species), which increases the family's genomic coverage by 68% and expands its phylogenetic diversity by 48.14%."

The method used to calculate Faith's PD was introduced into the Methods section. The phylogenetic reconstruction underlying this analysis is presented in Supp. Fig. S12. We believe this revision makes the scope of taxonomic and phylogenetic novelty explicit and fully addresses the reviewer's concern.

Comment 5: "*line 207-209 - while i am (tentatively) on board with the likening of ancestral genomic features based on short branch lengths within a ribosomal tree (though i think it would be safer to just stick to overall 'divergence', rather than speculating about specific features), inferring that modern lineages experience similar eco-evolutionary pressures to more ancient ones is too speculative, at least for the results section.*"

Response 5: We thank the reviewer for this careful observation. We fully agree that the original phrasing overreached in linking short branch lengths to specific ancestral genomic features, and in drawing parallels between ancient and contemporary eco-evolutionary pressures. These interpretations were speculative and not sufficiently supported by the available data.

In response, we have removed the relevant sentences and revised the paragraph to focus solely on empirical observations of branch lengths, positive selection, and genomic signatures within CSP1-4. The revised text now reads as follows (lines 294-318):

*"Phylogenetic analyses reveal that *Limnocylintranceae* shares a common ancestry with the CSP1-4 clade (Fig. 1a). The latter occupies an exceptionally short branch stemming from its most recent common ancestor—the shortest in our phylogeny—indicating reduced evolutionary change from the ancestral state (0.087 substitutions per site since divergence). To further understand the evolutionary landscape that may have favoured the emergence of *Limnocylintranceae*, we conducted clade-specific selection force analyses on a set of approximately 800 genes conserved across the four clades. We observed that CSP1-4 exhibits a 60% increase in the number of genes under diversifying (positive) selection compared to its ancestral lineage (i.e., QHBO01), along with a 25% increase in the number of sites under positive selection (Supp. Table S1). The elevated levels of positive selection observed in CSP1-4, the highest among the four clades, suggest that this lineage is undergoing adaptive evolution, characterized by an accumulation of non-synonymous substitutions. Additionally, the relaxation of selective pressure in CSP1-4 is complemented by an increased transposase density, a higher number of redundant proteins, and larger intergenic spacer regions—the longest among the four clades (Fig. 1b, Supp. Fig. S3). Together, these factors indicate decreased negative (purifying) selection and an enhanced potential for adaptive evolution^{15,33,34}. These observations align with the habitat expansion of CSP1-4 in sediment and aquatic ecosystems."*

Comment 6: "*line 223 - what is meant by 'align'? I think one could safely say correlate...*"

Reply 6: We thank the reviewer for this helpful clarification. In response, we have replaced "align" with "correlate" in line 317.

Comment 7: *"line 226 - again, it is unclear to me how methods based on modern genome sequences allow you to speak to selection pressures of the past."*

Reply 7: We thank the reviewer for this important point. As noted in our response to **Comment 5**, we have revised the relevant section to remove speculative interpretations about historical selection pressures. The updated text now focuses exclusively on observed genomic signatures in extant lineages, without extending inferences to ancestral conditions.

Comment 8: *"line 246 - is it possible that the acquired genes (or a subset of them) themselves explain the changes in GC? or is there evidence for GC remodeling more generally across the genome? this seems important to rule out among otherwise closely related species."*

Response 8: We agree with the reviewer that horizontally acquired genes—or a subset thereof—could plausibly contribute to the observed changes in GC content. To assess this, we compared the GC content of species-specific genes with that of the shared core genome. This analysis did not reveal substantial differences between the two categories, suggesting that the increase in GC content is not driven by the acquisition of genes with atypical base composition.

As this analysis did not yield additional insight beyond confirming genome-wide trends, we now present it in Supp. Fig. S13. We have also updated the main text to clarify that species-specific gene sets do not differ in GC content from conserved genes (lines 349-350): *"no substantial differences in GC content were observed between species-specific genes and the shared core genome."*

Comment 9: *"line 298 - is there any precedent for these curve shapes/plots in the literature? I do not think there really is a justification for connecting four discrete points with a trend line. if you must, then please visually indicate the points (is it just one per lineage per codon?)"*

Response 9: We thank the reviewer for this thoughtful observation and agree that our original presentation may have inadvertently implied a level of continuity not supported by the underlying data. Our intention was not to depict a formal statistical trend, but rather to illustrate a conceptual pattern—specifically, the funnel- or wine-bottle-shaped distributions frequently observed across *Limnocyndria* lineages, as shown in Supp. Figs. S6 and S7.

To address the reviewer's concern, we revised the figure (Fig.4) to explicitly display individual data points and replaced the continuous lines with dotted curves to minimize the risk of visual overinterpretation. Additionally, the figure legend has been updated to clarify that the plotted curves are conceptual representations, derived from the amino acid codon frequency distributions shown in Supp. Figs. S6 and S7 (lines 442-444): *"The dotted trendlines are conceptual representations, derived from the amino acid codon frequency distributions shown in Supp. Figs. S6 and S7."*

We believe these revisions improve clarity and more accurately convey the nature of the underlying data, and we thank the reviewer for their helpful suggestion.

Comment 10: *"line 386 - while the hypothesis advanced here is interesting, it feels incomplete. are there other repair systems that could compensate for the lack of dna*

glycosylases? it would seem important to rule this out before making the claim that this lineage is uniquely mutation-prone, and even further, that this mechanism may underlie genome-wide GC shifts, especially without supporting experimental evidence."

Response 10: We appreciate this critical point and agree that a robust mechanistic foundation is needed. In response, we have clarified the lesion-specific role of the missing DNA glycosylases and added comparative evidence from model systems and natural lineages where similar losses have been studied.

(i) Non-redundancy of base excision repair (BER) enzymes:

The lesions handled by Tag (K01246), AlkA (K01247) and MPG (K03652) (**small N-alkyl adducts**), family-4 UDG (TIGR03914), Ung (K03648) and Mug (K03649) (**uracil in DNA**), Nei (K05522) (**oxidised pyrimidines**) and the nucleotide-sanitising enzyme MutT (K03574) (**pre-emptive hydrolysis of 8-oxo-dGTP**) are all small, helix-stable modifications that are poor substrates for the general repair systems. Bacterial nucleotide-excision repair (UvrABC) excels at bulky distortions and can remove thymine glycol, but displays negligible activity toward 3-methyl-guanine, uracil, or uracil-containing mismatches. DNA mismatch repair corrects a subset of U·G mispairs yet cannot process U·A or excise damaged bases directly, and therefore still depends on a DNA glycosylase for final lesion removal. Direct-reversal enzymes such as AlkB oxidatively demethylate only a limited subset of alkyl lesions (e.g. 1-methyl-A, 3-methyl-C) and do not compensate for the broader specificity of Tag/AlkA/MPG. Homologous recombination and translesion synthesis promote survival at stalled forks but leave the original chemical modification intact, requiring subsequent base-excision repair. As the authors are aware, no other pathway—alone or in combination—restores the full spectrum of activities provided by the simultaneous presence of these eight BER genes, underscoring why their concerted loss cannot be fully compensated in bacterial genomes.

(ii) Empirical evidence for mutagenesis and GC erosion upon glycosylase loss:

- In *E. coli*, loss of the two 3-methyladenine DNA glycosylases Tag (K01246) and AlkA (K01247) produces an ≈ 10 - to 20-fold rise in the spontaneous mutation rate and pronounced hypersensitivity to methyl-methanesulphonate (MMS), consistent with unrepaired 3-methyl-guanine that yields G \rightarrow A transitions (Grzesiuk et al., 2001).
- Deletion of ung (K03648) increases G·C \rightarrow A·T transitions about 30-fold because uracil generated by cytosine deamination is left in the genome (Duncan & Weiss, 1982).
- A deficiency of the nucleotide-sanitising phosphohydrolase MutT (K03574) elevates overall mutation frequencies by 10^2 - 10^4 -fold, with a diagnostic A·T \rightarrow C·G transversion spectrum that stems from misincorporation of 8-oxo-dGTP (Bessman et al., 1996).
- Although the oxidised-pyrimidine glycosylase Nei (K05522) partially overlaps with EndoIII/Nth, nei single mutants (with nth still present) are still measurably more sensitive to H₂O₂ and γ -irradiation and show a modest (≈ 2 -3-fold) increase in GC \rightarrow TA transitions, indicating that lesions such as thymine glycol and 5-hydroxy-cytosine are not fully repaired (Jiang et al., 1997).

Together, these gene-specific deficiencies are all increasing mutagenic potential and in 3 cases eroding GC by the following $-G \rightarrow A$, $G \cdot C \rightarrow A \cdot T$, $A \cdot T \rightarrow C \cdot G$. Their documented magnitudes demonstrate that, even with Nth present, the simultaneous absence of Tag, AlkA, MPG (K03652), family-4 UDG (TIGR03914), Ung, Mug (K03649), Nei and MutT would leave no bacterial repair pathway capable of fully compensating for the resulting load of alkylated, deaminated and oxidised bases.

(iii) Relevance to *Limnocylintranceae*:

In this lineage, the simultaneous loss of DNA glycosylases that target the three principal lesion classes—alkylation, deamination, and oxidation—constitutes a rare convergence of vulnerabilities. The attendant rise in $G \rightarrow A$ and $G \rightarrow T$ substitutions, together with the markedly reduced GC content, extensive genome reduction, and accelerated sequence evolution, all point to a mutational landscape dominated by unrepaired base damage. Elevated mutation rate has repeatedly been linked to genome shrinkage in prokaryotes (Bourguignon et al., 2020), and experimental-evolution work with *Salmonella typhimurium* has shown that strains lacking these repair activities can, by virtue of their underlying mutational bias, very rapidly lose GC content (Lind & Andersson, 2008).

We have expanded the Discussion accordingly (lines 653-665):

“In *Limnocylintranceae*, the concerted loss of Tag-/AlkA-/MPG-mediated alkyl repair, UDG-driven uracil repair, Nei-directed oxidation repair, and MutT nucleotide sanitising strips the lineage of its chief defences against the three most pervasive small-base lesions. Because nucleotide-excision repair, mismatch repair, homologous recombination and AlkB-type dioxygenases cannot excise 3-methyl-guanine, uracil, thymine glycol or 8-oxo-dGTP with comparable efficiency, these lesions will likely persist into replication. Laboratory strains deprived of the same activities manifest 10- to 10⁴-fold mutator phenotypes and strongly GC-eroding spectra— $G \rightarrow A$ (tag alkA), $G \cdot C \rightarrow A \cdot T$ (ung) and $GC \rightarrow TA$ (nei) (Bessman et al., 1996; Duncan & Weiss, 1982; Grzesiuk et al., 2001; Jiang et al., 1997). Such biases furnish a direct mechanistic path to the lineage’s AT-enrichment and gene loss, echoing broad phylogenomic evidence that links elevated mutation pressure to bacterial genome reduction (Bourguignon et al., 2020) and the rapid GC erosion produced in BER-deficient *Salmonella typhimurium* evolution experiments (Lind & Andersson, 2008).”

References:

Bessman, M. J., Frick, D. N., & O’Handley, S. F. (1996). The MutT proteins or “Nudix” hydrolases, a family of versatile, widely distributed, “housecleaning” enzymes. *Journal of Biological Chemistry*, 271(41), 25059-25062. <https://doi.org/10.1074/jbc.271.41.25059>

Duncan, B. K., & Weiss, B. (1982). Specific mutator effects of ung (uracil-DNA glycosylase) mutations in *Escherichia coli*. *Journal of Bacteriology*, 151(2), 750-755. <https://doi.org/10.1128/JB.151.2.750-755.1982>

Grzesiuk, E., Gozdek, A., & Tudek, B. (2001). Contribution of *E. coli* AlkA, TagA glycosylases and UvrABC-excinuclease in MMS mutagenesis. *Mutation Research/Fundamental and Molecular Mechanisms of Mutagenesis*, 480-481, 77-84. [https://doi.org/10.1016/S0027-5107\(01\)00171-3](https://doi.org/10.1016/S0027-5107(01)00171-3)

Jiang, D., Hatahet, Z., Blaisdell, J. O., Melamede, R. J., & Wallace, S. S. (1997). *Escherichia coli* endonuclease VIII: Cloning, sequencing, and overexpression of the nei

structural gene and characterization of nei and nei nth mutants. *Journal of Bacteriology*, 179(11), 3773-3782. <https://doi.org/10.1128/JB.179.11.3773-3782.1997>; JOURNAL: JOURNAL: JB; WGROUP: STRING: PUBLICATION

Lind, P. A., & Andersson, D. I. (2008). Whole-genome mutational biases in bacteria. *Proceedings of the National Academy of Sciences of the United States of America*, 105(46), 17878-17883. https://doi.org/10.1073/PNAS.0804445105/SUPPL_FILE/0804445105SI.PDF

Comment 11: *"line 405-408. What is the purpose of the DAPI stained image ? Or of the part of the graph that shows presence/absence of bacteriorhodopsin? As far as I can tell, neither of these are referenced in the main text and thus are confusing (also panels are mislettered). please either address in the main text or delete."*

Response 11: We thank the reviewer for drawing attention to this oversight. We agree that these elements were confusing and insufficiently integrated into the main text. This issue likely arose during a reshuffling of content between the main manuscript and the Supplementary Material. In response, we have removed the bacteriorhodopsin presence/absence panel and the DAPI-stained microscopy image from the main figure. The microscopy image is now included in the Supplementary Material (Supp. Fig. S10), where it is briefly described. We have also corrected the panel lettering accordingly.

Comment 12: *"410-413. Unfortunately, this sort of speculation has no place in a results section. what supports your claim that elevated mutagenesis led to the loss of AA biosynthetic pathways? from my perspective, the development of metabolic interdependencies/interactions could just as easily explain this observation (see the CPR bacteria as an example). I would strongly consider moving these sentences to the discussion section and motivating this analysis differently."*

Response 12: We thank the reviewer for this valuable feedback and agree that the causal link between elevated mutagenesis and loss of amino acid biosynthesis was too speculative for the Results section. We have now restructured the text accordingly. In the revised Results (lines 538-549), we present the observed biosynthetic deficiencies in *Limnocylintranceae* descriptively and without interpretation as follows:

"Limnocylintranceae genomes are missing the core enzymes for six canonical essential-amino-acid pathways (Fig. 4a), indicating a substantial, though not complete, contraction of their biosynthetic repertoire. This partial auxotrophy likely heightens dependence on the external amino acids or peptide-rich organic matter. Freshwater lakes, which often contain dissolved amino acids and oligopeptides, provide a potential source of these essential substrates (Krempaska et al., 2021). However, it remains unclear the extent to which reliance on external amino acids shaped the proteomic composition of Limnocylintranceae."

Comment 13: *"497-99, 532-34. While an interesting hypothesis, again, I do not think you can confidently conclude this evolutionary trajectory based on the evidence presented. The authors seem to focus on a favored hypothesis here (loss of DNA glycoylase-> hypermutation/plasticity) but do not explore alternative hypotheses. what about the rest of the genome? What other metabolic differences are observed between the freshwater and soil/sediment clades? As the authors themselves note in the introduction, these changes too may help explain the ability of lineages to cross ecological boundaries. Without exploring these alternatives fully, the conclusion risks coming across as cherry-picked."*

Response 13: We thank the reviewer for this thoughtful and constructive comment. We agree that our previous framing placed disproportionate weight on the hypermutation hypothesis and underrepresented other mechanisms relevant to ecological adaptation. In response—building on **Comment 2** and integrating material from the Supplementary Discussion—we have revised the Discussion section to present a more balanced view of evolutionary trajectories within the *Limnocylintranceae* and *CSP1-4* clades. Specifically:

1. We have expanded our treatment of adaptive processes in *CSP1-4*, highlighting habitat-specific gene acquisitions, increased positive selection, and genome expansion as key features of their transition into sediment and aquatic environments.
2. We now present elevated mutagenesis in *Limnocylintranceae* not as the primary cause of ecological success, but as a **mechanism that may have accelerated genome shrinkage**, enabling more rapid loss of biosynthetic functions dispensable in freshwater environments.

This revised framing emphasizes that **niche-directed genome evolution** is likely the unifying theme, with *CSP1-4* and *Limnocylintranceae* representing two contrasting evolutionary states: one based on functional innovation and expansion, the other on mutational pruning and metabolic simplification. Ancestral state reconstructions (Supp. Fig. S14) reinforces this dichotomy, showing *CSP1-4* gains in flagellar assembly, chemotaxis and lysine biosynthesis, versus *Limnocylintranceae* gains of bacteriorhodopsins accompanied by losses of eight amino-acid biosynthetic pathways. We believe these revisions offer a more integrative and accurate interpretation of our findings. We have added the following paragraph to the discussion section (lines 637-657) :

“Ecological transitions in Limnocylintranceae and CSP1-4 exemplify niche-directed genome evolution²⁸, yet the sister clades appear to sit at opposite ends of that continuum. CSP1-4 presents genomic signatures consistent with diversifying selection accompanied by net gene acquisition. Soil representatives retain a broad plant-polymer catabolic arsenal; sediment-dwelling taxa carry additional nitrate-respiratory modules and microaerobic terminal oxidases; and freshwater representatives uniquely encode flagellar and chemotaxis loci—features uncommon in other Chloroflexota³³ (Supp. Data S12). By contrast, Limnocylintranceae shows only modest habitat-linked gene gains, most notably the acquisition of proton-pumping rhodopsins (Supp. Discussion).

This asymmetry may reflect divergent responses to relaxed selective constraint. In Limnocylintranceae, such relaxation appears to have permitted the accumulation of loss-of-function mutations in DNA repair genes—including Tag/AlkA/MPG, UDG-family, Nei, and MutT (Fig. 4a)—leading to the degradation of base excision repair (BER) pathways that defend against alkylated, deaminated, and oxidized bases. The resulting hypermutator state likely accelerated GC content erosion and contributed to the loss of multiple biosynthetic pathways (Fig. 5b). Thus, while CSP1-4 seems to have navigated habitat transitions through adaptive gene gain and genome expansion, Limnocylintranceae appears to have undergone reductive genome evolution and metabolic simplification under sustained mutation pressure following BER collapse.”

Comment 14: "510-516. Again, what evidence substantiates these claims about evolutionary dead ends? If there is literature you can cite in service of this argument, please do so, otherwise this is entirely speculative and should be written less assertively."

Response 14: We thank the reviewer for this important observation. We agree that the original wording was too assertive and suggestive of determinism. In response, we have revised this section of the text to frame the concept of evolutionary constraint more cautiously and to clarify that the "cul-de-sac" metaphor refers to a potential reduction in future adaptive flexibility, rather than a definitive evolutionary impasse.

We have also added citations to studies in microbial systems where genome reduction and hypermutator phenotypes have been associated with diminished long-term evolvability (e.g., *Prochlorococcus*, *Pseudomonas aeruginosa*), and now emphasize that such outcomes are context-dependent and probabilistic rather than certain.

*"Having shed a large part of its metabolic repertoire (Fig. 4bc) and accumulated extensive sequence changes (Fig. 2), Limnocyndraceae exhibits a highly reduced genomic architecture. While this streamlining likely facilitated niche specialization, the combination of genome shrinkage (Fig. 1b) and elevated mutation pressure (Fig. 2a) may also limit the lineage's capacity to explore new adaptive landscapes. Similar trade-offs have been documented in other microbial systems: in *Prochlorococcus*, genome reduction restricts physiological plasticity and niche range (Marais et al., 2008; Ulloa et al., 2021), while hypermutator strains of *Pseudomonas aeruginosa* lose fitness when introduced to new environments (Hall et al., 2022). Thus, although Limnocyndraceae remains evolutionarily viable, its current genomic configuration may reduce future evolutionary flexibility—a possible example of the tension between hyperadaptability and hyperspecialization (Wielgoss et al., 2013)."*

Comment 15: "l12 - maybe replace 'unparalleled' here. while of course dependent on bacteria, viruses are also present in nearly every ecosystem on earth as well."

Response 15: We thank the reviewer for this helpful suggestion. We have replaced "unparalleled" with "remarkable" to provide a more accurate and balanced description of bacterial adaptability.

Comment 16: "l17 - what is meant by 'integrative'? perhaps clearer to phrase as 'integrating X and Y?..'"

Response 16: We thank the reviewer for this helpful suggestion. In line with this and other related comments, we have revised the abstract to replace the term "integrative" with "comparative genomics", which more precisely reflects our methodological approach (lines 15-16).

Comment 17: "l133 - nice use of sandpiper here!"

Response 17: We thank the reviewer for the kind remark.

Comment 18: "l194 - why are sigma factors being singled out here, among all genomic features?"

Response 18: We thank the reviewer for this thoughtful question. Sigma factors are highlighted here because they are a well-established proxy for transcriptional regulatory complexity and ecological versatility. A reduction in sigma factor number often reflects

diminished regulatory plasticity and is associated with specialization to narrow or stable niches. This trend has been documented in multiple genome-reduced bacteria, including *Candidatus Pelagibacter ubique* and *Buchnera aphidicola*, which retain only two sigma factors compared to the broader sets found in their free-living relatives. The loss of alternative sigma factors frequently parallels the broader elimination of accessory and stress-response genes during genome reduction. In this context, we believe that including sigma factor counts is both informative and relevant to the broader patterns of metabolic streamlining and ecological adaptation discussed in the manuscript.

We added the following at lines 258-259: "...sigma factors (proxies for transcriptional regulatory complexity and ecological versatility)..."

REVIEWER 2

Comment 1: *"The research represents important evolutionary work addressing how microbial species are able to transition from one environment to others, a knowledge gap with few well-demonstrated examples in the literature. Chloroflexota is a good lineage with adapted lineages to a variety of environments, making it ideal to study to address associated questions. Overall, the methods are generally sound and the paper is well written."*

Response 1: We thank the reviewer for their thoughtful and encouraging comments. We are pleased that the evolutionary scope of our study, the selection of *Chloroflexota*, and the overall execution of the work were well received.

Comment 2: *"However, the manuscript does at times infer and speculate on mechanisms that drive the adaptability and survival of the lineage during ecosystem transitions, particularly in the results section. Specific examples are noted in the line specific comments, and I would recommend a revision to address some of the perhaps overly speculative correlations. Additionally, it may be helpful to incorporate the results of additional methods to uncover ancestral genes using ancestral state reconstructions. I would recommend either a rephrasing of some inferences in the results/discussion, or the demonstration of these processes with additional methods."*

Response 2: We thank the reviewer for this constructive and thoughtful feedback. We agree that some interpretations in the original manuscript may have been stated too strongly, particularly within the Results section. In response, we have carefully revised the relevant passages throughout the manuscript, rephrasing or softening inferences where they relied on indirect evidence. In addition, we have followed the reviewer's suggestion to incorporate additional methods, including ancestral state reconstruction (see Methods; lines 933-959), to better support claims related to habitat transitions and gene loss. We address each specific instance flagged in the line-by-line comments below and have revised the manuscript accordingly.

Comment 3: *"Throughout the manuscript, the terms non-adaptive vs adaptive selection is used. However, negative vs. positive selection, and purifying vs. diversifying selection are also terms that are used. Please forgive me if there was a purpose for all these different*

terms in particular usages, but it would help a reader if one set of terms was consistent throughout the manuscript."

Response 3: We thank the reviewer for this helpful observation. We agree that the use of multiple terms to describe selective processes (e.g., adaptive vs. non-adaptive, positive vs. negative, purifying vs. diversifying) may be confusing to readers. While we note that both positive (diversifying/Darwinian) and negative (purifying) selection can be adaptive depending on context, we appreciate the need for terminological clarity. To simplify and standardize our usage throughout the manuscript, we have adopted the terms **positive/negative** selection to describe the direction of selective pressure at the nucleotide/gene level, and **adaptive/non-adaptive processes** when referring to broader evolutionary frameworks. We believe this revision improves consistency and readability.

Comment 4: "Line 26: Related to Figure 2D - the GC content of the freshwater adapted clade is lower than its ancestors, is this truly "exceptionally high"?"

Response 4: We thank the reviewer for pointing this out. At line 26, our use of "exceptionally high" refers to the GC content of *Limnocylintranceae* (median 63.15%; n = 98 MAGs) in the context of its high GC content despite a highly reduced genome size (median estimated genome size ~1.31 Mbp).

We believe this is notable because low GC content is commonly associated with reduced genome size in free-living bacteria, making the high GC content of *Limnocylintranceae* (~1.31 Mbp genome) an exception to this general trend. For comparison:

- *Ca. Actinomarinales* (~1.1 Mbp; GC ~32–33%) (López-Pérez et al., 2020)
- *Ca. Pelagibacter ubique* HTCC7211 (1.46 Mbp; GC 30.8%) (Stingl et al., 2007)
- *Prochlorococcus marinus* MED4 (1.66 Mbp; GC 30.8%) (Dufresne et al., 2003)
- SAR86 (~1.0 Mbp; GC 33–34%) (Roda-Garcia et al., 2023)
- *Ca. Nitrosopelagicus brevis* CN25 (1.23 Mbp; GC 33.2%) (Santoro et al., 2015)
- OM43 clade (*Methylophilales* HIMB624; 1.33 Mbp; GC 35.4%) (Huggett et al., 2012)
- *Ca. Fonsibacter ubiqvis* (1.16 Mbp; GC 29.02%) (Henson et al., 2018)

Regarding Fig. 2d, we note that all *Limnocylintranceae* (the only genus of *Limnocylintranceae*) species depicted are freshwater-adapted. While some internal variation is evident (e.g., the basal species at 66.8% vs. others at ~63.1%), the key point is that all members of the genus maintain a GC content significantly higher than typically seen in genome-reduced bacteria.

To further clarify evolutionary patterns and address potential confusion, we have added ancestral state reconstructions of habitat (Supp. Fig. S14; main nodes Fig. 1a) and now annotate habitat affiliations for all genomes in Fig. 1a. Additionally, Supp. Fig. S9 plots genome size versus GC content for freshwater *Limnocylintranceae* in comparison with other genome-reduced bacterial lineages, providing broader context for the unusually high GC content observed in this family.

References:

López-Pérez, M., Haro-Moreno, J. M., Iranzo, J., & Rodríguez-Valera, F. (2020). Genomes of the " Candidatus Actinomarinales" Order: Highly Streamlined Marine Epipelagic Actinobacteria . *MSystems*, 5(6).

Stingl, U., Tripp, H. J., & Giovannoni, S. J. (2007). Improvements of high-throughput culturing yielded novel SAR11 strains and other abundant marine bacteria from the Oregon coast and the Bermuda Atlantic Time Series study site. *The ISME Journal*, 1(4), 361-371.

Dufresne, A., Salanoubat, M., Partensky, F., Artiguenave, F., Axmann, I. M., Barbe, V., Duprat, S., Galperin, M. Y., Koonin, E. V., Le Gall, F., Makarova, K. S., Ostrowski, M., Oztas, S., Robert, C., Rogozin, I. B., Scanlan, D. J., De Marsac, N. T., Weissenbach, J., Wincker, P., ... Hess, W. R. (2003). Genome sequence of the cyanobacterium *Prochlorococcus marinus* SS120, a nearly minimal oxyphototrophic genome. *Proceedings of the National Academy of Sciences of the United States of America*, 100(17), 10020-10025.

Roda-Garcia, J. J., Haro-Moreno, J. M., Rodriguez-Valera, F., Almagro-Moreno, S., & López-Pérez, M. (2023). Single-amplified genomes reveal most streamlined free-living marine bacteria. *Environmental Microbiology*, 25(6), 1136-1154.

Santoro, A. E., Dupont, C. L., Richter, R. A., Craig, M. T., Carini, P., McIlvin, M. R., Yang, Y., Orsi, W. D., Moran, D. M., & Saito, M. A. (2015). Genomic and proteomic characterization of "*Candidatus Nitrosopelagicus brevis*": An ammonia-oxidizing archaeon from the open ocean. *Proceedings of the National Academy of Sciences of the United States of America*, 112(4), 1173-1178.

Huggett, M. J., Hayakawa, D. H., & Rappé, M. S. (2012). Genome sequence of strain HIMB624, a cultured representative from the OM43 clade of marine Betaproteobacteria. *Standards in Genomic Sciences*, 6(1), 11-20.

Henson, M. W., Lanclos, V. C., Faircloth, B. C., & Thrash, J. C. (2018). Cultivation and genomics of the first freshwater SAR11 (LD12) isolate. *The ISME Journal*, 12(7), 1846.

Comment 5: "*Line 68-82: What about Actinobacteria? See work such as Ghai et al., 2013 (Metagenomics uncovers a new group of low GC and ultra-small marine Actinobacteria)*"

Response 5: We thank the reviewer for drawing our attention to the work by Ghai et al. (2013) and for highlighting the role of *Actinobacteria*. Based on Fig. 1b of that study, a close phylogenetic relationship is suggested between marine and freshwater clades within the *Acidimicrobiia*. To explore this further, we examined the habitat distribution of several orders within *Acidimicrobiia* as represented in the GTDB database (including *Actinomarinales*, ATN_3, MCC26256, UBA2766, and UBA5794). While *Actinomarinales* appears to be restricted to marine environments, other related clades are found across a range of aquatic and terrestrial habitats, including freshwater, sediments, and wastewater systems. Although the precise directionality of the habitat transition (freshwater-to-marine or vice versa) remains unclear, we have included *Actinomarinales* as a potential example of a freshwater-marine transition in the revised manuscript, denoted as "...*Actinomarinales* (freshwater to marine?)" to reflect this uncertainty (lines 84-85).

Comment 6: "*Line 129-132: Was this done with an outgroup?*"

Response 6: We thank the reviewer for this helpful observation. The phylogenomic reconstruction discussed in lines 129-132 was indeed rooted using an outgroup. To make this clearer, we have revised the text at line 127 to read: "*We built a rooted phylogenomic tree using maximum-likelihood methods...*". Additionally, while the legend of Fig. 1a already indicated that the tree is rooted, we have now made this more explicit by adding: "*Outgroup rooting was performed using members of the Eremiobacterota phylum.*"

Further details regarding phylogenetic reconstruction, including the rooting strategy, are provided in the Methods section under the subheading “Evolutionary history reconstructions.”

Comment 7: “Line 152: Figure 1 - the green colors transitioning from A to B are kind of confusing given the two sister clades are red and green in A, but only green in B. Maybe indicate both clades as green in A?”

Response 7: We thank the reviewer for this helpful suggestion. As advised, we have updated Fig. 1a so that both sister clades are now depicted in green, ensuring visual consistency with panel B. Additionally, we have enhanced Fig. 1a by including habitat affiliations at the tips of each branch and by annotating ancestral habitat states at major phylogenetic nodes. Full details of the ancestral state reconstruction are now provided in Supp. Fig. S14.

Comment 8: “Line 201-209: This section in particular seems speculative - could be tested with ancestral state reconstruction.”

Response 8: We agree with the reviewer that the original phrasing was speculative. In response, we have revised this section to remove unsupported inferences. Additionally, we performed an ancestral state reconstruction of key metabolic pathways, the results of which are now presented in Supp. Fig. S14. A detailed discussion of these findings is included in the Supplementary Discussion (lines 127-152 in the supplementary information).

Comment 9: “Line 242: What is species defined as? (ANI/AAI?)”

Response 9: We thank the reviewer for pointing this out. Species were defined as MAGs sharing >95% average nucleotide identity (ANI) and >70% conserved DNA, consistent with widely accepted thresholds for bacterial species delineation (Jain et al., 2018). We have now clarified this in the manuscript (lines 336-340), which reads:

*“Selection-pressure analyses of 52 MAGs representing four *Limnocylinus* species—defined by >95% average nucleotide identity and >70% conserved DNA—indicate that the basal lineage contains both fewer genes and sites per gene under negative selection than its evolutionary younger, lower-GC relatives (Fig. 2a,c).”*

Reference:

Jain, C., Rodriguez-R, L. M., Phillippy, A. M., Konstantinidis, K. T., & Aluru, S. (2018). High throughput ANI analysis of 90K prokaryotic genomes reveals clear species boundaries. *Nature Communications* 2018 9:1, 9(1), 1-8. <https://doi.org/10.1038/s41467-018-07641-9>

Comment 10: “Line 248: No changes in what?”

Response 10: We thank the reviewer for pointing out this ambiguity. By “no changes,” we were referring to genes for which no nucleotide substitutions were detected across the alignments—i.e., genes that appear fully conserved within the dataset and are inferred to be under strong purifying selection. To improve clarity, we have revised the sentence to explicitly state this in the manuscript (lines 343-348).

*"Thus, the approximately 3% decrease in GC content within *Limnocyclus* is associated with an increase in both the number of genes and the number of sites under negative selection, along with a reduction in the number of fully conserved (invariable) genes—defined here as genes with no detectable nucleotide substitutions across the alignment, consistent with strong purifying selection."*

Comment 11: "Line 251: Not able to tell what are synonymous substitutions from Figure 2, unless all of the data depicted in Figure 2 is from dS? If so, this needs to be stated in the caption."

Response 11: We thank the reviewer for this comment. The sentence has been revised to refer specifically to Fig. 2c, where genes under negative selection are used to indicate the predominance of synonymous substitutions (i.e., nucleotide changes that do not alter the encoded amino acid and are typically interpreted as markers of negative selection). We also replaced "purifying selection" with "negative selection" for consistency in terminology.

The sentence now reads as follows (lines 350-355): *"This indicates that negative selection affects a larger portion of the genome, while an elevated mutation rate introduces more sequence variation, as evidenced by the predominance of synonymous substitutions—nucleotide changes that do not alter the amino acid sequence and are typically retained under negative selection—in most genes (Fig. 2c)."*

Comment 12: "Line 257: Figure 2D - could these differences be due to noise (not true signal) due to undersampling? Also, the GC content of the freshwater adapted clade is still lower than its ancestors (see line comment on the abstract)."

Response 12: To provide further clarification, Fig. 2d illustrates GC content dynamics within *Limnocyclus*, the only currently known genus of the *Limnocyclus* family. Both the genus and family are freshwater-adapted, as more clearly indicated in the revised Fig. 1a.

One of the main purposes of Fig. 2d is to show that:

(i) GC content decreases within *Limnocyclus* lineages after genome reduction has occurred. All clades depicted in Fig. 2d are part of the genome-reduced, freshwater-adapted *Limnocyclus* family, and comparisons are made within this group.

(ii) The decrease in GC content is more pronounced in genic regions than in intergenic spacers, likely due to the higher mutational target size of genes. In other words, longer genic regions accumulate more GC-reducing mutations simply by probability.

Fig. 3 further supports this interpretation by showing that these GC-reducing mutations are enriched at third codon positions—typically synonymous sites—indicating that they accumulate without affecting protein function and likely contribute to the observed GC decrease (Fig. 2a,c further substantiate this by showing how the number of sites/genes and genes under negative selection—mutations that do not alter amino acid identity—are increasing in the 3 species with lower GC content).

To improve clarity for readers, we have revised the manuscript to explicitly label the taxonomic level ("genus", "family", "class") and the ecological context ("freshwater-adapted") when referring to *Limnocyclus* and *Limnocyclus*.

Additionally, to rule out the possibility that the pattern in Fig. 2d could be explained by differences in GC content between shared and species-specific genes, we conducted a targeted comparison. As no substantial differences were found, this result is now included in Supp. Fig. S13.

Finally, we note that our analysis is based on GC content measurements across approximately 36,000 genes and intergenic regions (from >10 MAGs per species), and therefore, we do not believe that undersampling is a significant concern in this case.

Comment 13: "*Line 282-285: This seems speculative - is higher GC content a consequence of small size? What is the mechanism there?*"

Response 13: We thank the reviewer for this insightful comment. Our intention was not to suggest that higher GC content is a direct consequence of small genome size, but rather to highlight an unexpected pattern in how GC content is distributed between coding and intergenic regions in freshwater-adapted *Limnocyndraceae*. As shown in Fig. 2d, the GC content decrease in this lineage is more pronounced in protein-coding regions than in intergenic spacers.

This is counterintuitive, as intergenic regions—generally being non-coding and under weaker selective pressure—are typically more susceptible to mutation-driven GC erosion (Ochman, 2003; Raghavan et al., 2012). This observation holds for the other clades: *CSP1-4*, *QHBO01*, and *P2-11E* (Supp. Fig. S5a).

However, in *Limnocyndraceae*, we observe that intergenic regions are both extremely short (median length: 12 bp, Supp. Fig. S4b) and higher in GC content than coding regions (median gene length: 810 bp). We interpret this pattern as a consequence of mutational pressure (Fig. 2a,c) acting more strongly on the larger coding regions, simply due to their greater sequence length and thus greater probability of accumulating GC-reducing mutations. This contrasts with patterns observed in *CSP1-4*, *QHBO01*, and *P2-11E* (Supp. Fig. S5a), where coding regions are not disproportionately affected in this way. To avoid confusion, we have revised the text accordingly and added the intergenic spacer length to support our explanation.

The text at lines 386–392 now reads as follows:

"By examining four representative species, we found that in freshwater-adapted Limnocyndraceae, intergenic regions—despite their short median length (12 bp)—consistently exhibited higher GC content than coding regions (Fig. 2d). This pattern may reflect a reduced likelihood of mutation accumulation in these shorter sequences, as their limited length presents a smaller mutational target compared to longer coding regions."

References:

Ochman, H. (2003). Neutral Mutations and Neutral Substitutions in Bacterial Genomes. *Molecular Biology and Evolution*, 20(12), 2091–2096.
<https://doi.org/10.1093/MOLBEV/MSG229>

Raghavan, R., Kelkar, Y. D., & Ochman, H. (2012). A selective force favoring increased G+C content in bacterial genes. *Proceedings of the National Academy of Sciences of the United States of America*, 109(36), 14504–14507.

Comment 14: *"Line 285: Consider discussing implications for "Increased mutation rate is linked to genome reduction in Prokaryotes" (Bourguignon et al., 2020) within this sentence."*

Response 14: We thank the reviewer for this valuable suggestion. In response, we have revised the relevant paragraph to more explicitly connect our findings with the framework proposed by Bourguignon et al. (2020). While that study does not specifically examine GC content, it establishes that genome reduction in prokaryotes is frequently linked to increased mutation rates. Our observation of GC content reduction in coding regions of *Limnocylintranceae* is consistent with this broader mutational model and suggests that base composition erosion may represent an additional outcome of the same underlying processes. We believe this framing better contextualizes our result within current evolutionary theory.

We have added the following paragraph to the manuscript (lines 401–406): *"Although GC content was not directly analyzed in Bourguignon et al. (2020), their finding that genome reduction in prokaryotes is frequently associated with elevated mutation rates supports a broader mutational framework. Our results align with this model and suggest that mutation-driven genome erosion may also influence base composition—manifesting as GC content reduction—in genome-reduced, free-living lineages such as Limnocylintranceae."*

Comment 15: *"Line 288-289: Again no mechanistic explanation here. "*

Response 15: We thank the reviewer for pointing this out. In response, we have revised the relevant section to better clarify the proposed mechanism. Specifically, we now explain that the observed GC reduction in *Limnocylintranceae* likely results from mutation-driven processes, where longer coding regions serve as larger mutational targets compared to short intergenic spacers. This asymmetry in sequence length may account for the disproportionate accumulation of GC-lowering substitutions in genes. We have updated the paragraph accordingly (lines 386-392) to more clearly present this mechanistic rationale.

Comment 16: *"Line 290-308: Ok I see the reasoning behind some of the mechanism for the GC content changes driving the cell shape, but the size link is still unclear in the previous paragraph."*

Response 16: We thank the reviewer for this comment. In the previous paragraph, our reference to "size" pertains to the relative lengths of coding (genic) and non-coding (intergenic) regions within the genome. As clarified in our response to **Comment 13**, we have revised the text to explicitly state this distinction and to better explain how (and where) the differing lengths influence the accumulation of GC-reducing mutations. We hope the updated version improves clarity and avoids further misunderstanding.

Comment 17: *"Line 389: Figure 4 - consider matching colors with figure 1."*

Response 17: We thank the reviewer for this helpful suggestion. We have updated Fig. 4 to align the color scheme with that used in Fig 1, improving visual consistency across the manuscript. We agree that this change enhances clarity and makes the figures easier to interpret.

Comment 18: *"Line 442-449, and throughout: This section in particular seems like it belongs in a discussion section, not results (brings in previous studies, and has a defined conclusion sentence) - consider revision of the text to separate results from discussion, or*

to have one combined results and discussion section. This applies to other sections throughout the results."

Response 18: We thank the reviewer for the observation. We opted for a combined Results and Discussion format, which allows interpretation alongside data presentation. In light of the comment, we reviewed the relevant sections to ensure all conclusions are directly supported by results and that speculative language has been minimized.

Comment 19: *"Line 450: Figure 5 - again consider matching colors with figure 1 and 4, and others."*

Response 19: We thank the reviewer for this consistent and helpful suggestion. In response, we have updated the color scheme in Fig. 5 to match that used in Fig. 1 and 4, ensuring visual consistency throughout the manuscript. We agree that this improves the overall clarity and cohesion of the figures.

Manuscript title: "**Deep-branching Chloroflexota lineages illuminate the eco-evolutionary foundation of cross-ecosystem colonization**".

REVIEWER 1

Comment 1: *"The abstract is still too strongly worded for what is essentially an evolutionary hypothesis that cannot / is not proven definitively in the text. I suggest phrasing as 'we postulate' or 'we speculate' that.. based on X or Y evidence. As written, it is still far too deterministic."*

Response 1: We thank the reviewer for this important point and agree that portions of the previous abstract overstated causal certainty for inferences that are necessarily indirect. In response, we have thoroughly revised the abstract to ensure that (i) mechanistic statements are explicitly framed as hypotheses consistent with the genomic evidence, (ii) deterministic or teleological wording is removed, and (iii) the overall take-home message is expressed in probabilistic terms, consistent with comparative genomics-based inference.

Concretely, we made the following changes:

1. Mechanistic inferences are now explicitly conditional and evidence-linked. For example, we replaced causal phrasing implying certainty (e.g., "dismantled... likely fostering... accelerated... enabled") with language that directly reflects what is supported by the genomic patterns, i.e., "coincides with," "is consistent with," and "may have." In the revised abstract, the DNA repair signal is framed as: "loss of key DNA glycosylases coincides with degradation of base excision repair and is consistent with a hypermutator state that may have accelerated genomic erosion...".
2. We avoided teleology and deterministic endpoints. We replaced trajectory language that implied a predetermined outcome with neutral evolutionary phrasing (e.g., "gave rise to"), and maintained probabilistic interpretation throughout, in line with our general approach elsewhere in the revision (context-dependent, non-deterministic framing).
3. We softened the take-home message to an inference rather than a demonstration, and clarified causal direction. The conclusion now reads: "our findings are consistent with mutation-driven genome reduction and proteome optimization acting in concert to support cross-ecosystem boundary crossing and freshwater specialization..." This replaces the previous "Our findings demonstrate..." framing

Comment 2: *"Please incorporate citations to recent reviews covering microbial habitat transitions, including from/to freshwater environments. You may find these interesting points of comparison for your own work:*

<https://doi.org/10.1111/1462-2920.16313>

<https://www.annualreviews.org/content/journals/10.1146/annurev-micro-041320-032304>"

Response 2: We thank the reviewer for this helpful suggestion to strengthen the manuscript's framing within recent syntheses on microbial habitat transitions.

A. Incorporation of the suggested *Environmental Microbiology* review

In response, we have added the review (<https://doi.org/10.1111/1462-2920.16313>) to the revised manuscript. It is now cited at lines 75 and 77.

B. *Annual Review of Microbiology* article already included

The second suggested review (<https://doi.org/10.1146/annurev-micro-041320-032304>) was already included in the submitted version and has been retained in the revision. It is cited at lines 56, 69, and 75.

REVIEWER 3

Comment 1: "Reviewer #3 (Remarks to the Author):

I am reviewing a revised version of the manuscript. Overall, the manuscript reads well, and I consider that the authors have satisfactorily addressed most of Reviewer-2's comments.

One point I think should be discussed further is whether the quality of the MAGs could have influenced estimates of gene loss or genome reduction. As short-read (Illumina) MAGs typically miss genes from the accessory genome, I wonder whether the absence of genes (e.g., DNA glycosylases) could have been overestimated in specific lineages. In addition, were these MAGs manually curated to ensure that all contigs belong to the MAGs?

Response 1: We thank the reviewer for these comments. We agree that short-read (Illumina) MAGs can be incomplete and may under-represent the flexible (accessory) genome, which can, in principle, affect presence/absence inference. We therefore designed our analyses to minimize the risk of overestimating gene loss or genome reduction and to interpret "absence" conservatively at the lineage level.

A. MAG quality control and contig-level curation

All MAGs included in downstream analyses were subjected to stringent quality filtering (completeness/contamination thresholds; see Methods). To further reduce the risk of misbinning, we performed bin-curation steps that evaluate contig consistency (e.g., coverage, sequence composition, and taxonomic signal) and remove incongruent contigs. Importantly, the pdCEL-derived genomes used here are curated within the pdCEL framework, and our analyses prioritize these curated bins.

B. Why the focal genes are unlikely to be "missed accessory" genes

The reviewer is correct that accessory genes can be missed in fragmented MAGs. However, the genes at the centre of this discussion (e.g., DNA glycosylases and other core DNA maintenance functions) are typically not pangenome-restricted traits in closely related bacteria; they are generally part of the conserved cellular toolkit. Conversely, genes in the pangenome are, by definition, present only in subsets of strains/subpopulations, meaning they are absent from a substantial fraction of individuals. Thus, even if one were to posit rare strain-specific occurrences (e.g., via occasional HGT), this would still support our key interpretation: these functions are not broadly conserved

and are depleted from the lineage's core gene repertoire, rather than being systematically present but missed by assembly.

Relatedly, no single genome type (MAG, SAG, or isolate) can "resolve" the accessory genome on its own—capturing pangenome variation requires population-level sampling across multiple representatives. In this context, MAGs are advantageous because they are reconstructed from environmental populations and, in practice, tend to reflect the majority genomic signal present in those populations rather than a single idiosyncratic isolate.

C. Replication and dense sampling argue against a MAG artifact

Our inference does not rely on isolated absences in one or a few genomes. We assembled genomes independently from multiple time points, habitats, and ecosystems, and we analysed a densely sampled dataset for Limnocylintridae: 67 pdCEL genomes (curated) plus 31 publicly available genomes (total n = 98). The relevant genes were not detected across these 98 independent representatives, making it unlikely that the pattern is driven by sporadic MAG incompleteness or assembly dropout in specific samples.

D. Conservative conclusion

Based on the consistent absence across a large, curated, and multi-source genome set, we infer that these genes are generally absent (i.e., not conserved) in the lineage. We explicitly acknowledge that we cannot exclude the possibility that individual strains could sporadically acquire such functions via HGT, but such rare occurrences would not change the central result that these genes are not part of the lineage's conserved genomic repertoire.

We have clarified these points in the revised manuscript's Methods section:

"Because short-read MAGs can under-recover parts of the flexible (accessory) genome, we applied conservative criteria for interpreting gene presence/absence. Specifically, we restricted downstream analyses to quality-filtered, curated MAGs and inferred lineage-level gene depletion only when a gene was consistently undetected across multiple independent MAGs, rather than from single-genome non-detections. Accordingly, "absence" is interpreted as loss from the conserved (core) gene repertoire at the lineage level, while rare strain-specific occurrences (e.g., via horizontal transfer) cannot be formally excluded."